# Disturbed retinoid metabolism upon loss of *rlbp1a* impairs cone function and leads to subretinal lipid deposits and photoreceptor degeneration in the zebrafish retina

Domino K Schlegel[1], Srinivasagan Ramkumar[2], Johannes von Lintig[2], Stephan CF Neuhauss[1]*

[1]Department of Molecular Life Sciences, University of Zurich, Zürich, Switzerland; [2]Department of Pharmacology, School of Medicine, Case Western Reserve University, Cleveland, United States

**Abstract** The *RLBP1* gene encodes the 36 kDa cellular retinaldehyde-binding protein, CRALBP, a soluble retinoid carrier, in the visual cycle of the eyes. Mutations in *RLBP1* are associated with recessively inherited clinical phenotypes, including Bothnia dystrophy, retinitis pigmentosa, retinitis punctata albescens, fundus albipunctatus, and Newfoundland rod–cone dystrophy. However, the etiology of these retinal disorders is not well understood. Here, we generated homologous zebrafish models to bridge this knowledge gap. Duplication of the *rlbp1* gene in zebrafish and cell-specific expression of the paralogs *rlbp1a* in the retinal pigment epithelium and *rlbp1b* in Müller glial cells allowed us to create intrinsically cell type-specific knockout fish lines. Using *rlbp1a* and *rlbp1b* single and double mutants, we investigated the pathological effects on visual function. Our analyses revealed that *rlbp1a* was essential for cone photoreceptor function and chromophore metabolism in the fish eyes. *rlbp1a*-mutant fish displayed reduced chromophore levels and attenuated cone photoreceptor responses to light stimuli. They accumulated 11-*cis* and all-*trans*-retinyl esters which displayed as enlarged lipid droplets in the RPE reminiscent of the subretinal yellow-white lesions in patients with *RLBP1* mutations. During aging, these fish developed retinal thinning and cone and rod photoreceptor dystrophy. In contrast, *rlbp1b* mutants did not display impaired vision. The double mutant essentially replicated the phenotype of the *rlbp1a* single mutant. Together, our study showed that the *rlbp1a* zebrafish mutant recapitulated many features of human blinding diseases caused by *RLBP1* mutations and provided novel insights into the pathways for chromophore regeneration of cone photoreceptors.

\*For correspondence:
stephan.neuhauss@mls.uzh.ch

**Competing interest:** The authors declare that no competing interests exist.

## Introduction

The gene *RLBP1* encodes the cellular retinaldehyde-binding protein (CRALBP), a soluble 36 kDa protein involved in the vertebrate visual cycle. CRALBP belongs to the CRAL-TRIO family of proteins and contains a CRAL-TRIO domain common to several lipid-binding proteins (*Panagabko et al., 2003*). In mammals, CRALBP is expressed in the retinal pigment epithelium (RPE) and Müller glial cells (MGCs). It is known to bind 11-*cis*-retinal (11cisRAL) in the RPE and both 11cisRAL and 11-*cis*-retinol (11cisROL) in the retina (*Saari et al., 1982*). It is also able to bind 9-*cis*-retinal but not 13-*cis*-retinal (*Saari and Bredberg, 1987*). CRALBP facilitates both the isomerohydrolase reaction mediated by RPE65 and the oxidation of the product, 11cisROL, to 11cisRAL (*Saari et al., 2001*; *Saari et al., 1994*;

*Stecher et al., 1999*; *Winston and Rando, 1998*). Also, CRALBP prevents unwanted side reactions or premature photoisomerization of the unstable chromophore 11cisRAL (*McBee et al., 2001*).

Mutations in *RLBP1* have been associated with a group of autosomal recessive retinal diseases comprising autosomal recessive retinitis pigmentosa (*Maw et al., 1997*), Bothnia dystrophy (*Burstedt et al., 1999*), Newfoundland rod–cone dystrophy (*Eichers et al., 2002*), retinitis punctata albescens (RPA) (*Morimura et al., 1999*), and fundus albipunctatus (*Katsanis et al., 2001*). These retinal disorders differ in certain aspects of the disease such as time of onset, severity, and progression. However, they all cause prolonged dark adaptation and night blindness in early childhood. Upon progression, the involvement of the macula and RPE is possible, resulting in loss of central vision and legal blindness at the age of 20–40 (*Burstedt and Golovleva, 2010*). Another interesting common feature in patients with *RLBP1*-associated disease is a characteristic fundus appearance with homogeneously distributed, punctate white-yellow deposits. It was found that impaired CRALBP function can be caused by pathogenic mutations that either tighten or weaken the interaction of CRALBP with 11cisRAL (*Golovleva et al., 2003*). The crystal structure of wild-type and mutant CRALBP provided insights into the accompanying molecular changes within the protein (*He et al., 2009*).

The involvement of cone photoreceptors at later stages of the disease makes it imperative to elucidate the role of CRALBP in visual pigment regeneration. While visual pigment regeneration in rods is very well established (e.g., reviewed in *Kiser et al., 2014*), the processes involved in 11cisRAL recycling in cones are less well understood. Besides the canonical visual cycle, a Müller glia cell-dependent intraretinal visual cycle has been proposed (*Fleisch et al., 2008*; *Collery et al., 2008*; *Wang et al., 2009*; *Xue et al., 2015*; *Sato and Kefalov, 2016*) as well as recycling pathways via light-dependent mechanisms both in the RPE and MGCs (*Kolesnikov et al., 2021*; *Morshedian et al., 2019*; *Kaylor et al., 2017*; *Babino et al., 2015*). Together, these pathways might meet the demand for chromophore supply of cones during bright daylight illumination and prevent competition between cone and rod photoreceptor for the unique chromophore.

Interestingly, mammalian *RLBP1*, the gene encoding CRALBP, is expressed not only in RPE cells but also in Müller glia cells (MGCs) (*Bunt-Milam and Saari, 1983*). Therefore, CRALBP was implicated in the pathways for visual pigment regeneration in both cell types. Findings from morpholino-knockdown in zebrafish larvae supported this idea (*Fleisch et al., 2008*; *Collery et al., 2008*). However, the exact mechanisms underlying these complementary pathways and the role of CRALBP therein are still debated (*Kolesnikov et al., 2021*; *Ward et al., 2020*; *Zhang et al., 2019*; *Morshedian et al., 2019*; *Kiser et al., 2019*; *Kiser et al., 2018*).

In contrast to mammals, the zebrafish eyes harbor duplicates of Cralbp and express *rlbp1a* in RPE cells and *rlbp1b* in MGCs in a nonoverlapping pattern (*Fleisch et al., 2008*; *Collery et al., 2008*). In the current study, we exploited the fish to target Cralbp in an intrinsically cell-specific manner and investigated the contributions of the RPE and the MGC to cone visual pigment regeneration. The cone-dominant retina of the zebrafish and the practically exclusively cone-driven retinal responses in larval stages (*Branchek, 1984*; *Bilotta et al., 2001*) facilitated the investigation of cone-specific defects upon knockout of either *rlbp1* paralog in larvae up to 5 days postfertilization (dpf). Moreover, our model has the advantage over previous morpholino-mediated knockdown studies in that we could assess the progression of the disease beyond 5 dpf. Importantly, our fish model recapitulated the key characteristics of the human disease, including progressive retinal thinning and the appearance of subretinal lipid deposits, resembling the yellow-white dots in patient fundi.

## Results
### Generation of intrinsically cell-specific KO lines using CRISPR/Cas9

The nonoverlapping expression pattern of *rlbp1a* and *rlbp1b* in the zebrafish retina allowed us to target each Cralbp separately and to create intrinsically cell-specific knockout (KO) lines. The *rlbp1a*−/− line harbors a 19 bp deletion in exon four causing a frameshift and an early stop codon. The *rlbp1b*−/− line displays an in-frame 18 bp deletion in exon 4 (see Methods Table 2 for sequences). Both KOs were confirmed in retinal sections using immunohistochemistry with a CRALBP antiserum (*Figure 1*). In wild types, CRALBP immunofluorescence is present in the RPE cells and the MGCs. In *rlbp1a*−/−, the staining is only present in MGCs, while in *rlbp1b*−/−, staining is restricted to the RPE. Double-KO animals were generated by crossing the two single-mutant lines. The loss of Cralbp

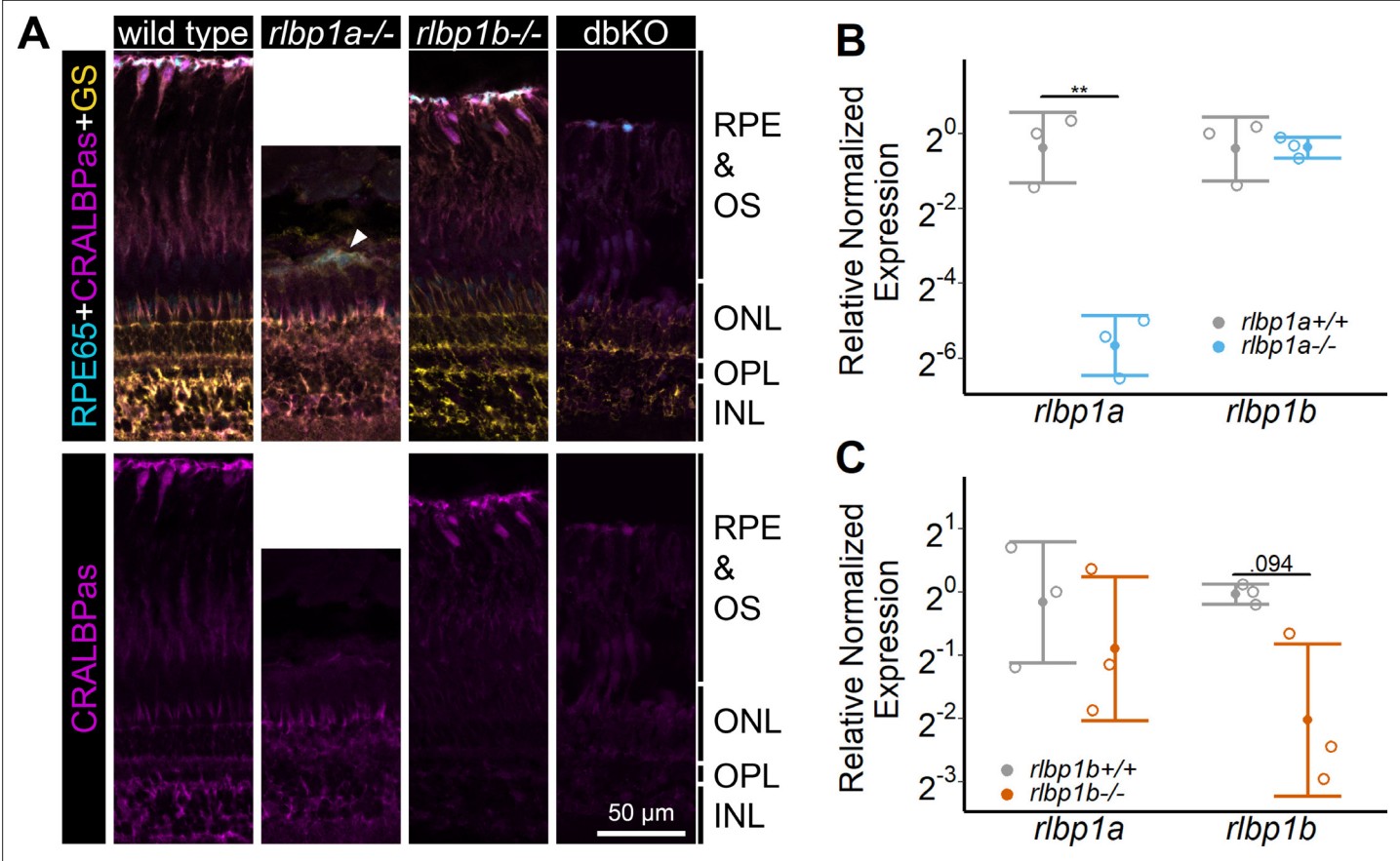

**Figure 1.** Protein and gene expression in Cralbp-knockout (KO) lines. (**A**) Immunohistochemical staining with cellular retinaldehyde-binding protein (CRALBP) antiserum (CRALBPas, magenta) and RPE65 (cyan) and glutamine synthetase (GS, yellow) as counterstain for RPE and Müller glia, respectively, on cryosections of adult retinas. In *rlbp1a−/−*, the layer containing RPE and outersegments is reduced in thickness and very disturbed, the white arrow indicates a remaining RPE65-positive cell. Bottom row showing CRALBPas channel, only. (**B, C**) Relative normalized expression of *rlbp1a* and *rlbp1b* in the eyes of the respective single-KO lines. Individual samples are shown with mean ± standard deviation (SD). (**B**) *rlbp1a* expression was reduced in *rlbp1a−/−* eyes [$t(4) = 7.39$, $p = 0.003$], while *rlbp1b* mRNA levels were comparable to wild type [$t(4) = −0.06$, $p = 0.955$]. (**C**) In *rlbp1b−/−*, *rlbp1a* expression was similar to wild type [$t(4) = 0.849$, $p = 0.444$], but *rlbp1b* mRNA levels were twofold reduced [$t(4) = 2.84$, $p = 0.094$]. Statistics: *t*-test comparing KO vs. wild type for each target with Benjamini–Hochberg correction for multiple testing. Significance: \*\*$p < 0.01$. Abbreviations: double-knockout (dbKO), retinal pigment epithelium (RPE), outer segment (OS), outer nuclear layer (ONL), outer plexiform layer (OPL), and inner nuclear layer (INL). See *Figure 1—source data 1* for expression data.

The online version of this article includes the following source data for figure 1:

**Source data 1.** Source data for *Figure 1*.

proteins in the retina of double-KO was confirmed by immunohistochemistry. Both single-KO lines had reduced levels of mRNA of the targeted gene, indicative of nonsense-mediated decay (*Figure 1B and C*). Notably, the expression of the respective paralogs was not increased. Therefore, transcriptional adaptation, often observed in zebrafish knockout models (*Rossi et al., 2015*), through upregulation of the paralogs, was not observed for Cralbps.

## Chromophore regeneration through 11-*cis*-retinyl esters is dependent on Cralbpa

To assess the effect of *rlbp1a* knockout on ocular retinoids, we determined their concentrations and composition in adult eyecups and 5 dpf whole larval extracts of mutants and controls. The term 11cisRAL always refers to protein-bound (e.g., by opsin or Cralbp) and free 11cisRAL in our experiments. Representative high-performance liquid chromatography (HPLC) traces are shown in *Figure 2—figure supplement 1*.

Adult eyecups were collected under ambient light. Dark-adapted (DA) larvae were divided into several groups (*n* = 100 larvae per group) that were subjected to different light conditions: DA, ambient light (ambient), 30 min bleach at 20'000 lux (30 min bright light [BL]), 1 hr bleach at 20'000 lux (1 hr BL), 1 hr bleach, and 1 hr redark adaptation (1 h reDA).

Chromophore levels in heterozygotes were comparable to wild types (*Figure 2A*, top panel). In *rlbp1a*−/− adults, 11cisRAL concentration was about threefold reduced when compared to wild-type siblings (*Figure 2A*, top panel). In DA *rlbp1a*−/− larvae, 11cisRAL levels were halved relative to wild type (*Figure 2B*, top panel). In line with previous reports (*Babino et al., 2015*), wild-type larvae maintained high levels of 11cisRAL even upon prolonged exposure to BL (*Figure 2B*, top panel). After one hour of redark adaptation, wild-type larvae regenerated the chromophore to levels of dark-adapted animals. In *rlbp1a*−/− larvae, ambient illumination caused a steep decline of 11cisRAL when compared to controls. After 1 hr of BL bleach, 11cisRAL levels were further reduced. Upon redark adaption, *rlbp1a*−/− larvae regenerated 11cisRAL to only 57 % of the level observed in dark-adapted larvae.

We also determined the concentration of 11cisREs which are Rpe65 dependently produced in the fish and serve as a reservoir for chromophore precursors under photopic conditions (*Babino et al., 2015*). Eyes of adult *rlbp1a*−/− fish displayed significantly higher 11cisRE concentrations than their wild type and heterozygous siblings (*Figure 2A*, middle panel). We observed a similar trend in larval samples. However, the effect was not as pronounced as in the adult eye (*Figure 2B*, middle panel). Illumination with BL led to a decrease in 11cisRE in both wild type and *rlbp1a*−/− larvae (*Figure 2B*, middle panel). Based on this observation and the reduced chromophore concentration in rlbp1a−/− larvae we assume that the esters were hydrolyzed upon light exposure but were not utilized for 11cisRAL production. Instead, we observed an increased concentration of atRE in *rlbp1a*−/− adult fish (*Figure 2A*, bottom panel) and a slight increase upon light exposure in *rlbp1a*−/− larvae (*Figure 2B*, bottom panel).

Together, our biochemical analyses showed that *rlbp1a*−/− adult and larval fish display a significant reduction of 11cisRAL concentration under all light conditions and an accumulation of 11-*cis* and atRE.

## Effects of Cralbpb on chromophore regeneration

Expressed in MGCs, *rlbp1b* was implicated as a critical component of the proposed intraretinal visual cycle (*Xue et al., 2015*; *Sato et al., 2017*). Therefore, we investigated putative effects brought up by the *rlbp1b* knockout on ocular retinoid concentrations and composition. We detected comparable amounts of 11cisRAL in *rlbp1b*−/−, *rlbp1b*+/−, and wild-type siblings (*Figure 2C*, top panel). The eyes of the *rlbp1b*−/− and *rlbp1b*±fish displayed lower 11cisRE and atRE levels than the wild-type control siblings (*Figure 2C*, middle and bottom panel).

We next challenged larval eyes with BL illumination. Subsequently, we extracted retinoids and subjected them to HPLC analysis. We observed that genetic disruption of *rlbp1b* did not alter 11cisRAL concentrations in DA, bleached, and bleached and redark-adapted *rlbp1b*−/− larvae when compared to controls (*Figure 2D*, top panel). Also, retinyl ester levels at this stage were not significantly altered (*Figure 2D*, middle and bottom panel).

In adult double mutant fish, we found a reduction in 11cisRAL levels (*Figure 2E*, top panel). Additionally, these fish displayed high levels of RE (*Figure 2E*, middle and bottom panel).

In conclusion, *rlbp1b* deficiency did not affect ocular retinoid concentration in such a dramatic fashion as observed in *rlbp1a*-deficient eyes. Additionally, the double mutant displayed decreased 11cisRAL and elevated RE concentrations when compared to wild-type controls. This pattern of alterations in ocular retinoid content was comparable to that found in *rlbp1a*−/− fish.

## Cone-specific retinal function in larvae is dependent on Cralbpa

To investigate the contribution of the two *Rlbp1* genes on cone photoreceptor responses, we measured DA white-light electroretinogram (ERG) in 5 dpf larvae. At this developmental stage, ERG responses are exclusively cone driven in zebrafish (*Branchek, 1984*; *Bilotta et al., 2001*). As a readout for retinal function, we used the b-wave amplitude generated by ON-bipolar cells. Maximum amplitude was assessed after stimulation with increasing light intensities within a range of five log units. Representative traces can be found in *Figure 3—figure supplement 1* .

We found a significant effect of genotype on b-wave amplitude in the *rlbp1a*-mutant line. In *rlbp1a*−/− larvae, amplitudes were significantly reduced when compared to heterozygous and

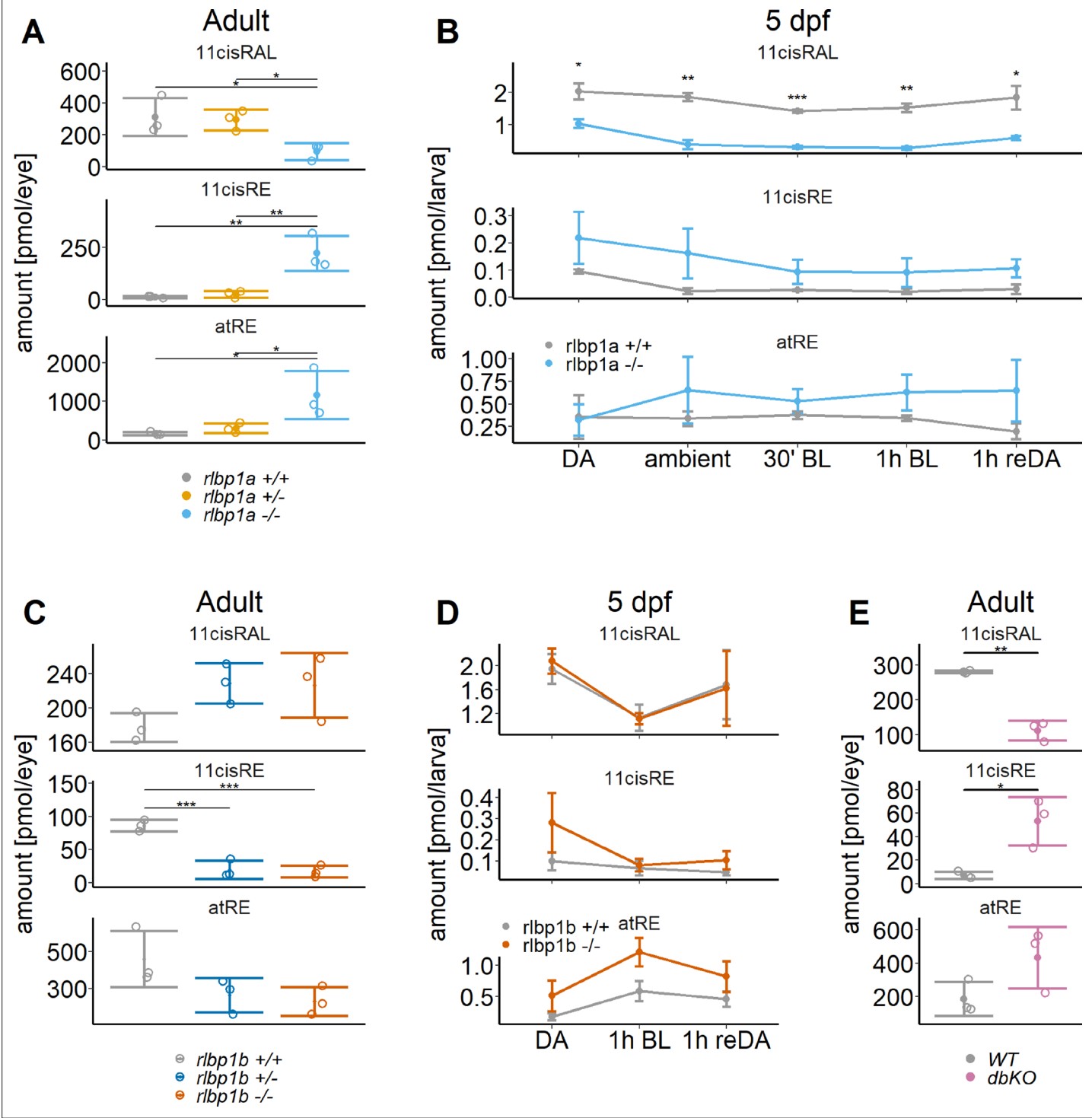

**Figure 2.** Ocular retinoid content. Retinoid content of adult eyes (**A, C, E**) or whole larvae (**B, D**), determined by high-performance liquid chromatography (HPLC) analysis. (**A, C, E**) Adults: samples were collected in ambient light. For each knockout (KO) line, 11cisRAL, 11cisRE, and atRE levels were determined. Data from *n* = 3 individual samples per genotype are shown with their respective mean ± standard deviation (SD). (**A**) *rlbp1a*-KO line: one-way analysis of variance (ANOVA) revealed a significant effect of genotype on the amount of retinoids for 11cisRAL [$F(2,6)$ = 6.215, p = 0.035], 11cisRE [$F(2,6)$ = 17.328, p = 0.003], and atRE [$F(2,6)$ = 6.526, p = 0.031]. (**C**) *rlbp1b*-KO line: genotype had an effect on 11cisRE levels (one-way ANOVA) [$F(2,6)$ = 40.774, p < 0.001], but less so on 11cisRAL [$F(2,6)$ = 3.378, p = 0.104] and atRE [$F(2,6)$ = 3.676, p = 0.091]. (**E**) double-KO line. 11cisRAL is reduced in dbKO [$t(4)$ = 10.2, p = 0.0005]. The retinyl esters 11cisRE [$t(4)$ = −3.84, p = 0.018] and atRE [$t(4)$ = −2.04, p = 0.112] accumulate (Student's *t*-test). (**B, D**) Larvae: single-KO larvae (*n* = 3 samples per genotype and condition, where one sample equals 100 larvae per condition and genotype) were

*Figure 2 continued on next page*

*Figure 2 continued*

collected after exposure to different light conditions (DA: dark-adapted; ambient: ambient light; 30' BL: 30 min exposure to bleaching light at 20'000 lux; 1 hr BL: 1 hr exposure to 20'000 lux; 1 hr reDA: 1 hr at 20'000 lux after wich the larvae were allowed to redark adapt for 1 hr). Samples were collected in three independent experiments per KO line and data for 11cisRAL, 11cisRE, and atRE levels are shown as mean ± SD and the main effect of genotype on the amount of retinoids was calculated with two-way ANOVA. (**B**) In *rlbp1a−/−* larvae11cisRAL levels are reduced [$F_{(1,28)} = 85.51$, $p < 0.001$] whereas retinyl esters 11cisRE [$F_{(1,28)} = 9.84$, $p = 0.004$] and atRE [$F_{(1,28)} = 4.133$, $p = 0.052$] are slightly elevated. (**D**) 11cisRAL regeneration in *rlbp1b−/−* larvae is not affected [$F_{(1,16)} = 0.003$, $p = 0.96$] and 11cisRE levels are comparable to rlbp1b+/+ larvae [$F_{(1,16)} = 2.246$, $p = 0.153$]. The amount of atRE is slightly elevated [$F_{(1,16)} = 6.05$, $p = 0.026$]. Pairwise comparisons to the respective control were performed with Benjamini–Hochberg correction for multiple testing. Significance levels are indicated by asterisks. $p \geq 0.05$ is not shown. *$p < 0.05$, **$p < 0.01$, ***$p < 0.001$. See *Figure 2—source data 1* for retinoid measurements.

The online version of this article includes the following source data and figure supplement(s) for figure 2:

**Source data 1.** Source data for *Figure 2*.

**Figure supplement 1.** High-performance liquid chromatography (HPLC) analysis of the retinoid composition of adult zebrafish eyes.

wild-type siblings (*Figure 3A*). Knockout of *rlbp1b* did not affect ERG responses under the applied conditions (*Figure 3B*).

To explore the functional consequences of Cralbpa deficiency during prolonged photopic illumination, we measured light-adapted ERG responses with constant background illumination (LA) (*Figure 4A*, left panel). In addition, we determined cone responses directly after a 1 hr bleach with BL (*Figure 4A*, middle panel). These conditions paralleled our retinoid analysis protocol. To investigate recovery of *rlbp1a−/−* larvae after such an intense bleach, we redark-adapted larvae for 1 hr after the bleach (reDA) (*Figure 4A*, right panel). The prolonged illumination reduced the sensitivity in mutant and wild-type larvae when compared to the DA state. For comparison, the dotted line represents the mean DA b-wave amplitude of wild types at 0.01 % light intensity.

The responses of *rlbp1a−/−* were diminished under all light conditions when compared to wild types. The sensitivity was shifted approximately one log unit and the maximum amplitude remained below wild-type levels in all illumination conditions examined. Notably, the responses of cone photoreceptors were not completely absent in the mutant, even after exposure to bright bleaching light. Also, we observed substantial recovery of retinal function following redark adaptation, albeit with reduced sensitivity.

This prompted us to investigate the possibility that *rlbp1b* compensates for the lack of *rlbp1a*. Therefore, we repeated the experiments with double-KO larvae. Again, we found a one log unit shift in sensitivity and reduced maximum amplitudes in DA double-KO responses (*Figure 3C*). Also under photopic conditions (*Figure 4B*), the phenotype was similar to the one seen in *rlbp1a−/−*. However, immediately after a strong bleach, but also following redark adaptation, the sensitivity shifted more than one log unit in double-KO compared to the respective wild-type controls. Interestingly, responses were severely attenuated but still recordable at 100 % light intensity in the bleach condition.

Together, our analyses demonstrated that cone function depends on RPE-expressed Cralbpa. We observed no pronounced changes of cone responses in DA Cralbpb-deficient larval eyes and only a minor additive effect in the double mutant under the applied conditions.

## Knockout of RPE-expressed Cralbpa leads to subretinal lipid deposits and age-dependent photoreceptor degeneration

Histological examination of *rlbp1a⁻ᐟ⁻* larvae at 5 dpf revealed an overall normal stratification of the different retinal layers (*Figure 5A and B*). In the ventral retinal periphery, we found fewer rod outer segments (*Figure 5*, black arrows), indicating either slower development or an early loss of rod outer segments (OSs). This observation is further supported by the decrease in *rhodopsin* mRNA levels at this age *Figure 6A*. The thickness of the outer nuclear layer (ONL), containing the photoreceptor nuclei, was unchanged (*Figure 5G1*). However, already at this developmental stage, round subretinal lesions became detectable that appeared colorless upon staining with Richardson's solution (*Figure 5B2 and H1*). This phenotype became more apparent at 34 dpf (*Figure 5C, D and H2*) and in 6- month-old fish (*Figure 5E, F and H3*). The lesions appeared to increase not only in number but also in size. Occasionally, we detected these structures in wild-type siblings (*Figure 5C2*). However, these tended to be smaller and sparser.

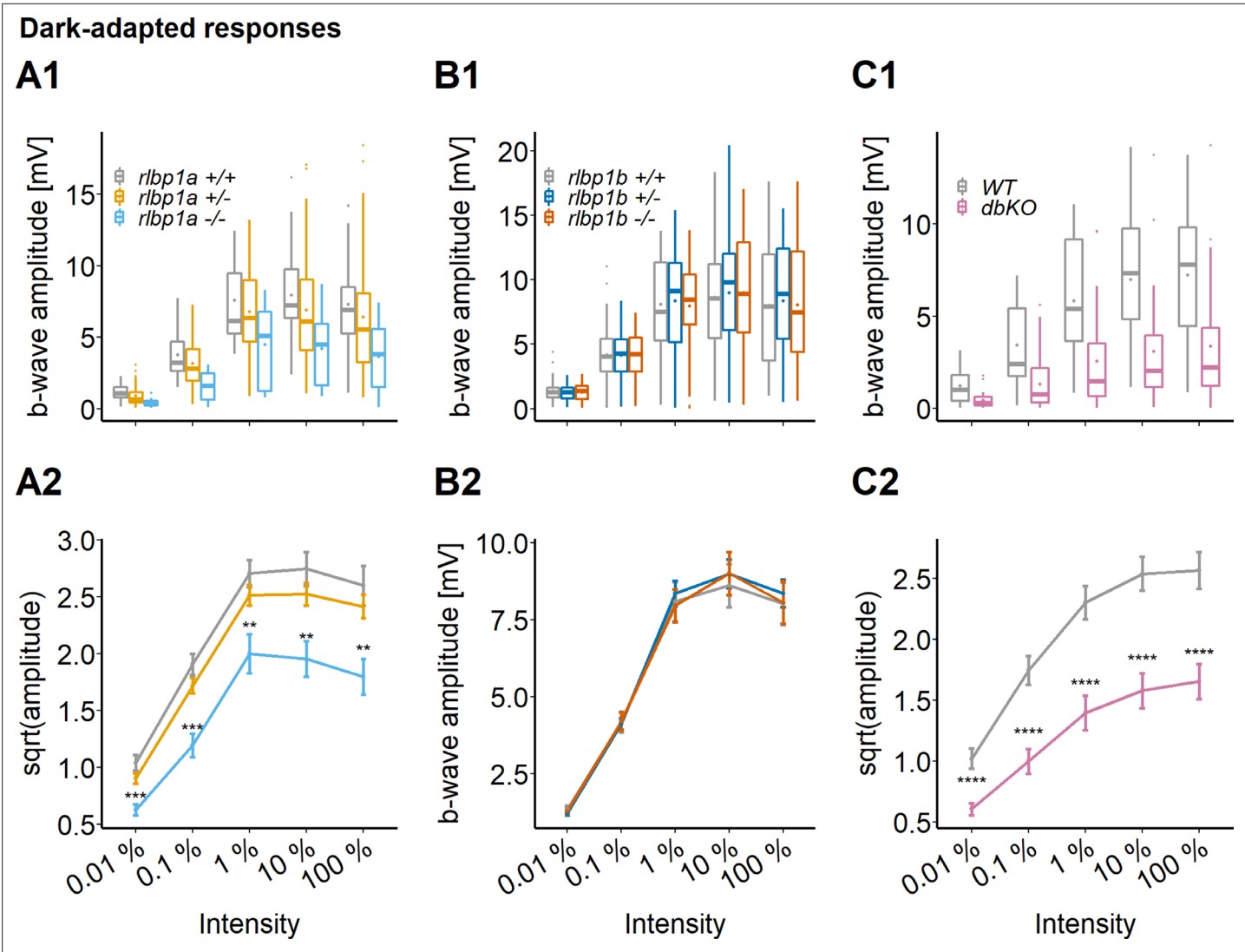

**Figure 3.** Cone-specific retinal function, dark-adapted responses. ERG responses at 5 days postfertilization (dpf). Responses were probed in the different knockout (KO) lines at five increasing light intensities under dark-adapted conditions. (**A1–C1**) Data are shown as boxplots of the b-wave amplitude with the box depicting the interquartile range (IQR), whiskers are 1.5 × IQR, mean is shown as diamond, median as horizontal line in the box and outliers are points outside of 1.5 × IQR. Data were collected in $N \geq 3$ independent experiments and from total $n \geq 18$ larvae per genotype (A: *rlbp1a*-KO line) or $n \geq 43$ larvae per genotype (B: *rlbp1b*-KO line) or $n \geq 29$ larvae per genotype (C: double-KO line). (**A2–C2**) Data are shown as mean ± standard error of the mean (SEM). (**A**) Respones of the rlbp1a-KO line. (**A2**) Repeated-measures analysis of variance (ANOVA) revealed a significant effect for genotype [$F_{(2,435)} = 33.649$, $p < 0.001$]. (**B**) Respones of the *rlbp1b*-KO line. (**B2**) Repeated-measures ANOVA did not reveal any significant effect of genotype on amplitude [$F_{(2,915)} = 0.185$, $p = 0.83$]. (**C**) Responses of double-KO and wild types. (**C2**) Repeated-measures ANOVA revealed a significant effect for genotype [$F_{(1,290)} = 99.808$, $p < 0.001$]. Statistical analysis was performed on data in (**A2, C2**) after square root transformation to mitigate the impact of outliers. Pairwise comparisons between wild type (WT) and mutants at each light intensity were computed and corrected for multiple testing using the Benjamini–Hochberg correction. Significance levels of pairwise comparisons depicted as asterisks: **$p < 0.01$, ***$p < 0.001$, ****$p < 0.0001$. See *Figure 3—source data 1* for b-wave amplitudes.

The online version of this article includes the following source data and figure supplement(s) for figure 3:

**Source data 1.** Source data for *Figure 3*.

**Figure supplement 1.** Exemplary electroretinogram (ERG) traces.

Importantly, we also found that already at 34 dpf, the OSs of both rods and cones were severely shortened, dysmorphic, or sometimes completely absent (*Figure 5D*). The double-KO (*Figure 7*) largely recapitulated the findings in *rlbp1a−/−* fish. In adults, it appeared that rod outer segments were slightly better preserved in double-KO than in the *rlbp1a−/−* single mutant (compare *Figures 5F, 7F*). Analysis of *rhodopsin* mRNA expression revealed lower expression at 5 dpf and in adults in

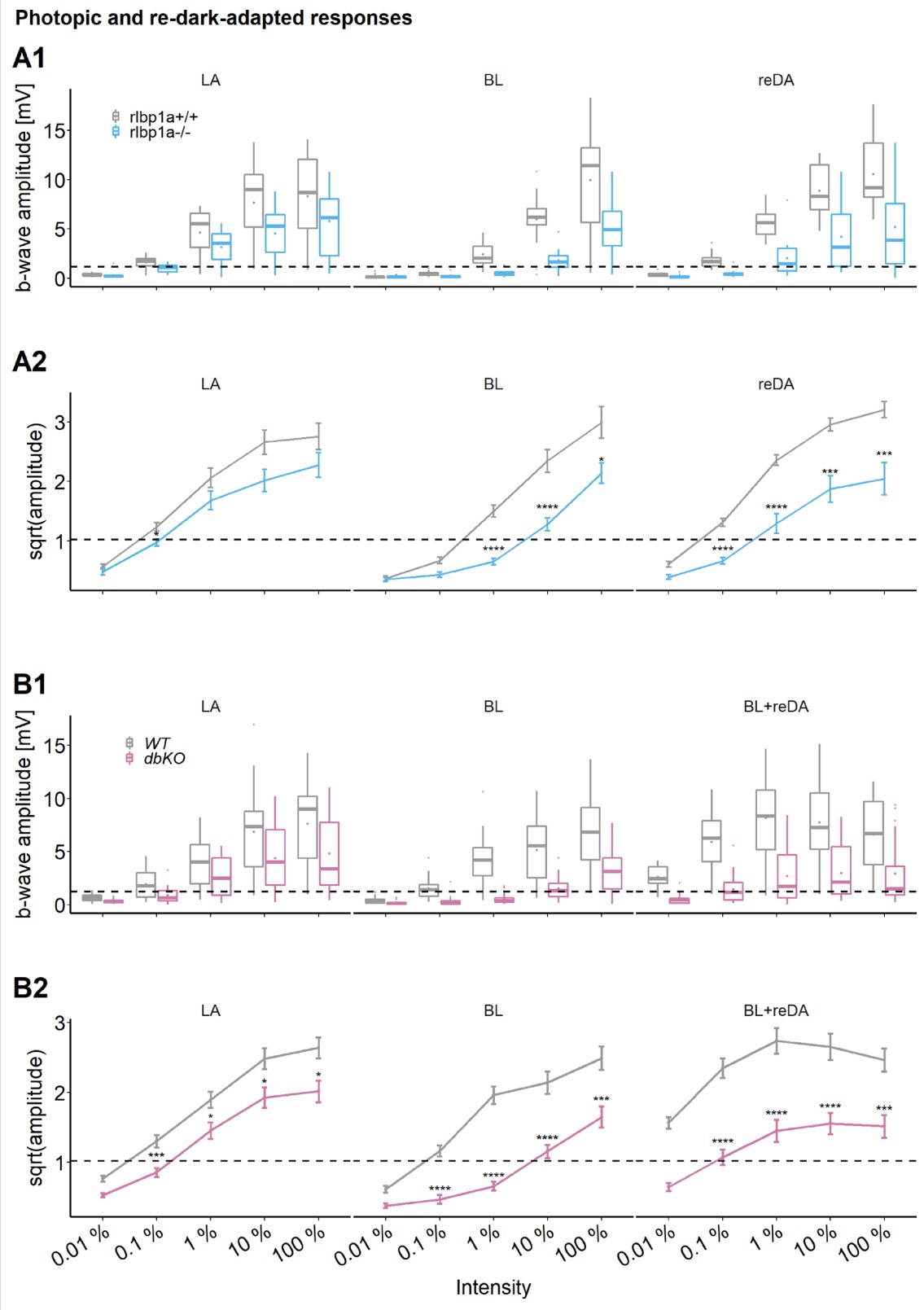

**Figure 4.** Cone-specific retinal function, photopic and redark-adapted responses. Photopic and redark-adapted electroretinogram (ERG) responses at 5 days postfertilization (dpf). (**A1, B1**) Data are shown as boxplots of the b-wave amplitude with the box depicting the interquartile range (IQR), whiskers are 1.5 × IQR, mean is shown as diamond, median as horizontal line in the box and outliers are points outside of 1.5 × IQR. Responses were probed at five increasing light intensities and under different background illumination. LA: light-adapted, no dark adaptation before measurements

*Figure 4 continued on next page*

*Figure 4 continued*

and background illumination during recordings. BL: recordings immediately after 1 hr exposure to bleaching light at 20'000 lux. reDA: recordings after bleaching and 1 hr redark adaptation. Dotted lines depict the mean responses of wild-type controls after dark adaptation at 0.01 % light intensity to illustrate the shift in sensitivity under photopic conditions. Data were collected in $N \geq 3$ independent experiments and from total $n = 15$ larvae per genotype and light condition (A: *rlbp1a*-knockout [KO] line), or $n \geq 20$ larvae per genotype and light condition (B: double-KO line). (**A2, B2**) Data are shown as mean ± standard error of the mean (SEM). (**A2**) Left panel: light-adapted responses of *rlbp1a−/−* vs. *rlbp1a+/+*. Repeated-measures analysis of variance (ANOVA) revealed a significant effect for genotype on b-wave amplitude [$F_{(1,112)} = 13.476$, $p < 0.001$]. Middle panel: responses after a bleach. Effect of genotype: [$F_{(1,84)} = 47.908$, $p < 0.001$]. Right panel: responses after bleach and redark adaptation. Effect of genotype: [$F_{(1,112)} = 81.267$, $p < 0.001$]. (**B2**) Left panel: light-adapted responses of dbKO vs. wild type (WT). Effect of genotype: [$F_{(1,236)} = 32.633$, $p < 0.001$]. Middle panel: responses after a bleach. Effect of genotype: [$F_{(1,172)} = 125.858$, $p < 0.001$]. Right panel: responses after bleach and redark adaptation. Effect of genotype: [$F_{(1,168)} = 106.625$, $p < 0.001$]. Statistical analysis was performed on data in (**A2, B2**) after square root transformation to mitigate the impact of outliers. Only intensity levels where wild-type values were above the dark-adapted threshold were considered in the analysis. Pairwise comparisons between WT and mutants at each light intensity were computed and corrected for multiple testing using the Benjamini–Hochberg correction. Significance levels of pairwise comparisons depicted as asterisks: *p < 0.05, ***p < 0.001, ****p < 0.0001. See *Figure 4—source data 1* for b-wave amplitudes.

The online version of this article includes the following source data for figure 4:

**Source data 1.** Source data for *Figure 4*.

both *rlbp1a−/−* single and double-KO. However, the reduction of *rhodopsin* mRNA expression in *rlbp1a−/−* adults was ~ fourfold whereas in the double-KO it was roughly twofold *Figure 8B,D*. Also, even though the number of white lesions was comparable to the single-KO, they appeared slightly smaller in the double-KO. Notably, these subretinal lesions were rare (*Figure 8*) and ONL thickness was maintained (*Figure 8E*) in *rlbp1b+/+* and *rlbp1b−/−* fish.

RPE cells form lipid droplet-like structures called retinosomes (*Imanishi et al., 2004a*; *Imanishi et al., 2004b*; *Palczewska et al., 2010*; *Orban et al., 2011*). They serve as storage sites for retinyl esters and can be visualized with lipophilic dyes, similar to other lipid droplets. The high concentration of RE in *rlbp1a−/−* and the location of the lesions between photoreceptor OS and the RPE layer, led us to investigate the possibility that the observed structures were enlarged retinosomes in the RPE.

To test this, we stained cryosections with BODIPY TR methyl ester, a lipophilic dye that stains neutral lipids and endomembranous structures like mitochondria and photoreceptor OS. Indeed, we found an increased amount of very bright spherical structures positive for BODIPY staining in the subretinal space in *rlbp1a−/−* and double-KO retinas (*Figures 5 and 7*). These droplets were rarely present in wild types (*Figures 5, 7, 8G*) and *rlbp1b−/−* fish (*Figure 8G*).

## Discussion

Our results showed that cones in the zebrafish retina rely on RPE-expressed Cralbpa for chromophore regeneration and retinal function. We found that the knockout of *rlbp1a* impaired cone-driven ERG responses and led to disturbed OS morphology of both rods and cones and age-dependent thinning of the photoreceptor nuclear layer. HPLC analysis of *rlbp1a−/−* revealed an accumulation of retinyl esters and a reduction in 11cisRAL that was associated with this visual impairment. To our knowledge, we present the first animal model that recapitulates both the age-dependent retinal thinning and the appearance of subretinal lesions that are commonly found in patients suffering from *RLBP1* mutations.

Our data indicate that in early stages, the lack of 11cisRAL (*Figure 2A, B and E*, top panel) is most likely the primary cause for the reduction in cone-mediated retinal function in the *rlbpa1−/−* fish. However, at later stages, the disturbed OS morphology and the loss of photoreceptors are likely to contribute to a deterioration of vision in these fish. In addition, it seemed that the integrity of both rods and cones is affected in *rlbp1a−/−* and double-KO fish. The elevated retinyl ester levels in *rlbp1a−/−* and double-KO zebrafish (*Figure 2A, B and E*, middle and lower panel) display as enlarged retinosomes found upon histological examination. ONL thinning appeared, concomitant with the increased number of enlarged retinosomes, only with age. It is interesting to note, that chromophore regeneration was not completely blocked but only attenuated in Cralbpa deficiency. This was true not only for *rlbp1a−/−*, but also for the double-KO.

These findings are consistent with the prolonged dark adaptation in patients with Bothnia dystrophy (*Burstedt et al., 2003*), and in mouse models (*Saari et al., 2001*; *Lima de Carvalho et al., 2020*). Previously, *Stecher et al., 1999* proposed that CRALBP facilitates the hydrolysis of 11cisRE. More recently, it was reported that the 11cisRE pool in the zebrafish RPE is mobilized in a light-dependent

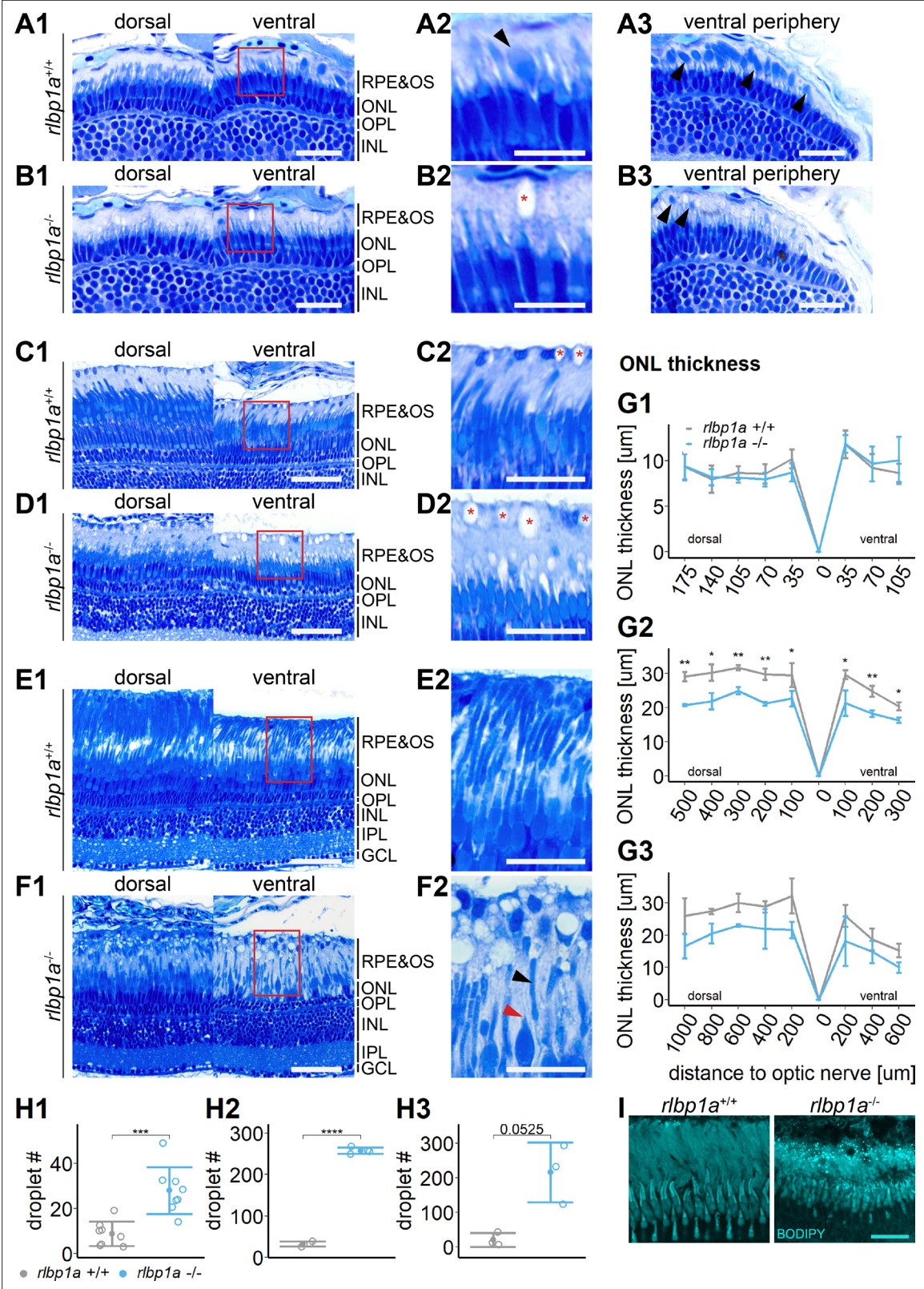

**Figure 5.** Accumulation of subretinal lipid deposits, dysmorphic outer segments, and age-dependent retinal thinning in *rlbp1a*-knockout (KO). (**A1–F1**) Central retinal semithin plastic sections (dorsal and ventral of the optic nerve) of *rlbp1a+/+* and *rlbp1a−/−* 5 dpf larvae (**A, B**), 1- month-old juveniles (**C, D**), and adults (**E, F**). (**A2–F2**) Closeup of area in red box. Subretinal lipid deposits in *rlbp1a−/−* retinas are found already at 5 days postfertilization (dpf) and increase in number and size with age. Occasionally, these structures are found in wild-type controls, but are usually smaller in size. Black

*Figure 5 continued on next page*

*Figure 5 continued*

arrows: rod outer segments. Red star: subretinal lipid deposits. Red arrow: missing cone outer segment. (**A3, B3**) Ventral peripheral retina. Black arrows indicate rod outer segments. These are less abundant, but not completely absent in KO retinas. (**G**) Quantification of outer nuclear layer (ONL) thickness central retinal sections in 5 dpf (**G1**), 1- month-old (**G2**), and adult (**G3**) retinas. Data are from n ≥ 3 biological replicates per genotype and age. Shown is the mean ± standard deviation (SD) at specified distances from the optic nerve (dorsal and ventral). Effect of genotype on ONL thickness (repeated-measures analysis of variance [ANOVA]) at 5 dpf (**G1**) [$F_{(1,16)} = 0.029$, $p = 0.87$], at 1 month (**G2**) [$F_{(1,4)} = 144.518$, $p < 0.001$], and in adults (**G3**) [$F_{(1,4)} = 18.642$, $p = 0.012$]. Pairwise comparisons of *rlbp1a−/−* vs. *rlbp1a+/+* by region were calculated with Benjamini–Hochberg correction for multiple comparisons. (**H**) Quantification of the number of lipid droplets in 5 dpf (**H1**), 1- month-old (**H2**), and adult (**H3**) retinas. Comparison of *rlbp1a−/−* vs. *rlbp1a+/+* was performed using *t*-test: (**H1**) [$t_{(10.5)} = -4.61$, $p < 0.001$], (**H2**) [$t_{(4.87)} = -43.2$, $p < 0.001$], and (**H3**) [$t_{(2.21)} = -3.83$, $p = 0.0525$]. Individual data points (open circles) from $n ≥ 3$ biological replicates per genotype and age are shown with mean ± SD. (**I**) Staining for neutral lipids with BODIPY TR methyl ester confirms disturbed outer segment morphology and accumulation of lipid droplets in the subretinal space of *rlbp1a−/−* retina. The example shows adult retinas of the respective genotype. Significance levels depicted as asterisks: *$p < 0.05$, **$p < 0.01$, ***$p < 0.001$, ****$p < 0.0001$. Scale bars: (**A1, A3, B1, B3**) 20 µm, (**A2, B2**) 10 µm, (**C1–F1**) 50 µm, (**C2–F2**) 25 µm, and (**I**) 10 µm. See *Figure 5—source data 1* for measurements in G and H.

The online version of this article includes the following source data for figure 5:

**Source data 1.** Source data for *Figure 5*.

manner to meet the high demand for chromophore under bright daylight conditions (*Babino et al., 2015*). We now demonstrated that the mobilization of the 11cisRE pool is Cralbpa dependent and plays a critical role for cone visual pigment regeneration. Interestingly, we found that 11cisRE concentrations also declined in the absence of Cralbpa upon light exposure. This result raises the possibility that 11cisRE hydrolysis can occur Cralbpa independently. Thus, Cralbpa may be mainly required to assist the oxidation of 11cisROL to 11cisRAL, as earlier proposed by *Saari et al., 1994*. Additionally, Cralbpa may act as a chaperone for chromophores in the RPE and prevent the light-dependent isomerization of the released 11-*cis*-ROL diastereomer to the *trans*-diastereomer of retinol in the RPE. The latter would be reesterified by LRAT (lecithin retinol acyltransferase), an enzyme that has a high capacity for RE production. This reaction sequence would explain the increase in atRE concentration in illuminated *rlbp1a−/−* larvae.

As an acceptor of 11-*cis*-retinoids RPE-expressed Cralbpa is found at a convergence point for potentially all chromophore producing pathways of the RPE. Besides its well-known function in the classical visual cycle downstream of RPE65 and its role in the light-dependent 11cisRE mobilization described in this study CRALBP was also shown to support photic chromophore regeneration by RGR in bovine RPE (*Zhang et al., 2019*). On the other hand, knockout of MGC-expressed *rlbp1b* in zebrafish did not affect 11cisRAL content, nor morphology and only marginally affected retinal function in the double-KO in BL conditions. This observation excludes that the MGC-expressed Cralbp carries out the same function in chromophore regeneration as its counterpart in the RPE. Accordingly, it was previously shown that 11cisRE was mostly present in retinosomes of the RPE (*Babino et al., 2015*). However, we do not close out that Cralbpb may play different roles in the visual cycle. For instance, *Kiser et al., 2018* suggested that CRALBP in MGCs could serve as a chromophore storage site by binding 11-*cis*-retinoids. Also, RGR in MGCs was implicated in light-dependent chromophore regeneration (*Morshedian et al., 2019*) and Cralbpb may drive this reaction during BL conditions. In this line of thought our measurements of 11cisRAL in the rlbp1b-KO could be masked by RGR-bound retinoids and provide an explanation for the more severe phenotype in double-KO, despite the apparently normal amounts of 11cisRAL in rlbp1b-KO. Further research is needed to explore the role of Cralbpb in the zebrafish retina.

Early work in mice failed to reproduce the progressive loss of photoreceptors as well as the characteristic fundus findings seen in patients (*Saari et al., 2001*). A more recent study (*Lima de Carvalho et al., 2020*) reported ONL thinning in *Rlbp1−/−* mice between 8 and 12 months of age. Such changes may have gone unnoticed in earlier studies on *Rlbp1−/−* mice due to the younger age of animals at examination (5.5 months). Also in patients, the onset of degenerative processes can vary between individuals. Our zebrafish model recapitulates the early impairment of rods and subsequent cone dysfunction that culminates in photoreceptor degeneration at a relatively early age. Therefore, it resembles more closely an early-onset form of CRALBP-associated diseases, such as RPA or Bothnia dystrophy.

The white-yellow fundus lesions in patients with CRALBP-associated disease are usually found upon imaging the central retina, including the macular region and optic disc (*Burstedt and*

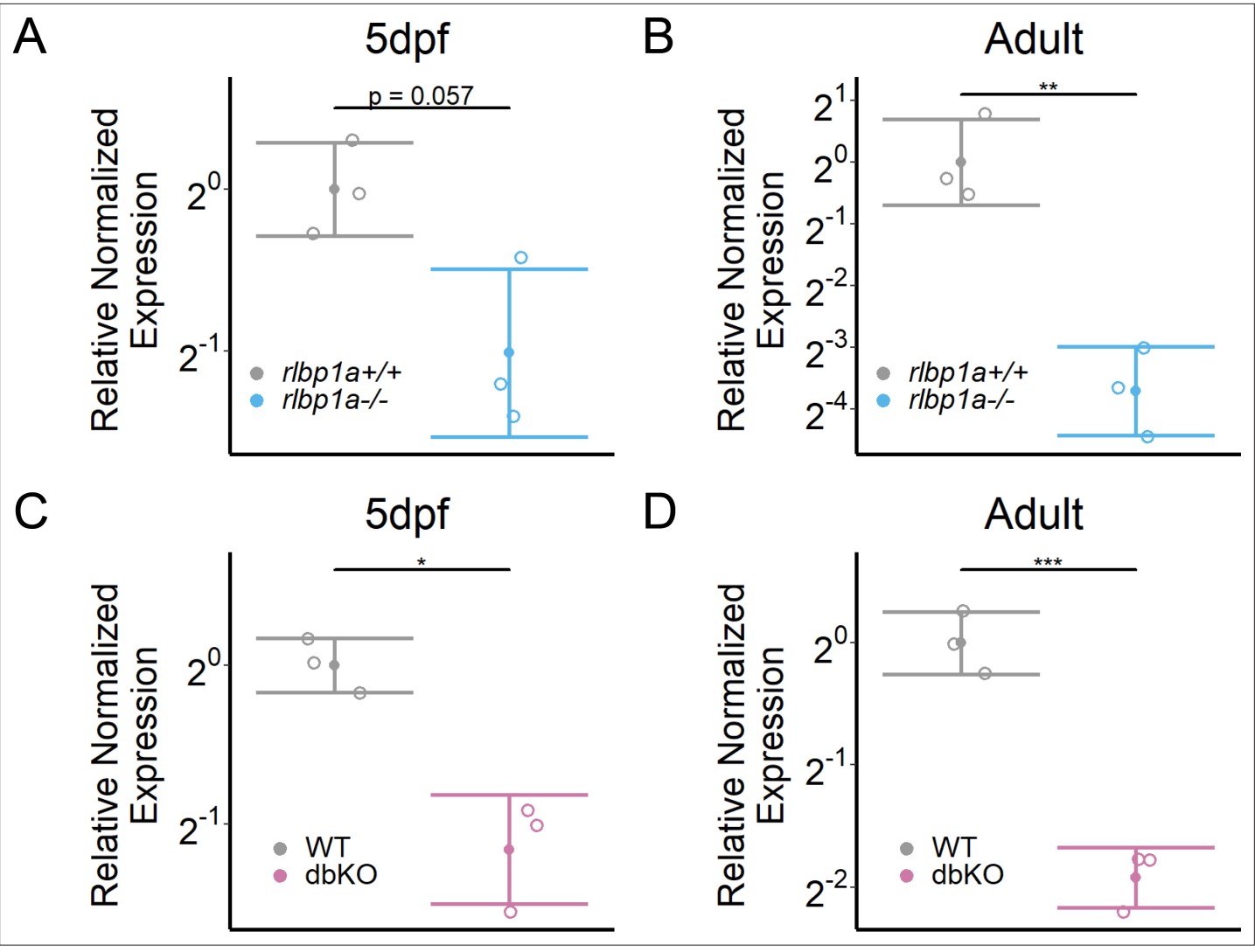

**Figure 6.** *rhodopsin* mRNA expression. Relative normalized expression of rhodopsin in the eyes of the respective knockout (KO) lines. Rhodopsin mRNA levels in the *rlbp1a*-KO line are reduced at (**A**) 5 days postfertilization (dpf) [*t*(3.13) = 2.95, p = 0.057] and (**B**) in adults [*t*(4) = 6.43, p = 0.003]. (**C**) Similar to the *rlbp1a*-KO, rhodopsin mRNA levels are reduced in double-KO at 5 dpf [*t*(2.93) = 5.22, p = 0.015] and (**D**) adults [*t*(3.99) = 9.35, p < 0.001]. Individual samples (open circles) are shown with mean ± standard deviation (SD). Statistics: *t*-test comparing double-KO vs. wild type. *p < 0.05, **p < 0.01, ***p < .001. See *Figure 6—source data 1* for expression data.

The online version of this article includes the following source data for figure 6:

**Source data 1.** Source data for *Figure 6*.

Golovleva, 2010). The lesions appear to be restricted in size towards the center. Interestingly, the white dots are often reported to be absent in the macula (*Demirci et al., 2004*; *Naz et al., 2011*) or at least the fovea (*Dessalces et al., 2013*). The white dots are not present at an early age (*Gränse et al., 2001*) and only become prominent in the teens and early adulthood, and slowly disappear with age (*Burstedt et al., 2001*). The macular sparing suggests that these structures are more likely to be associated with rod photoreceptors. Based on the localization of these structures between the photoreceptor and RPE layer upon OCT imaging and the fact that other visual cycle defects (e.g., mutations in *RDH5*, *RPE65*) result in a similar punctate fundus appearance, we suggest that they could represent retinyl ester-filled, enlarged retinosomes, similar to the ones we found in the zebrafish retina. If this and their association with rod photoreceptors, as mentioned above, holds true, it is curious, that despite an accumulation of retinyl esters, *Rlbp1−/−* mice were not reported to have enlarged retinosomes or similar structures. So, why does the cone-dominant zebrafish produce this phenotype? First, considering cone density, the larval retina nicely mimics

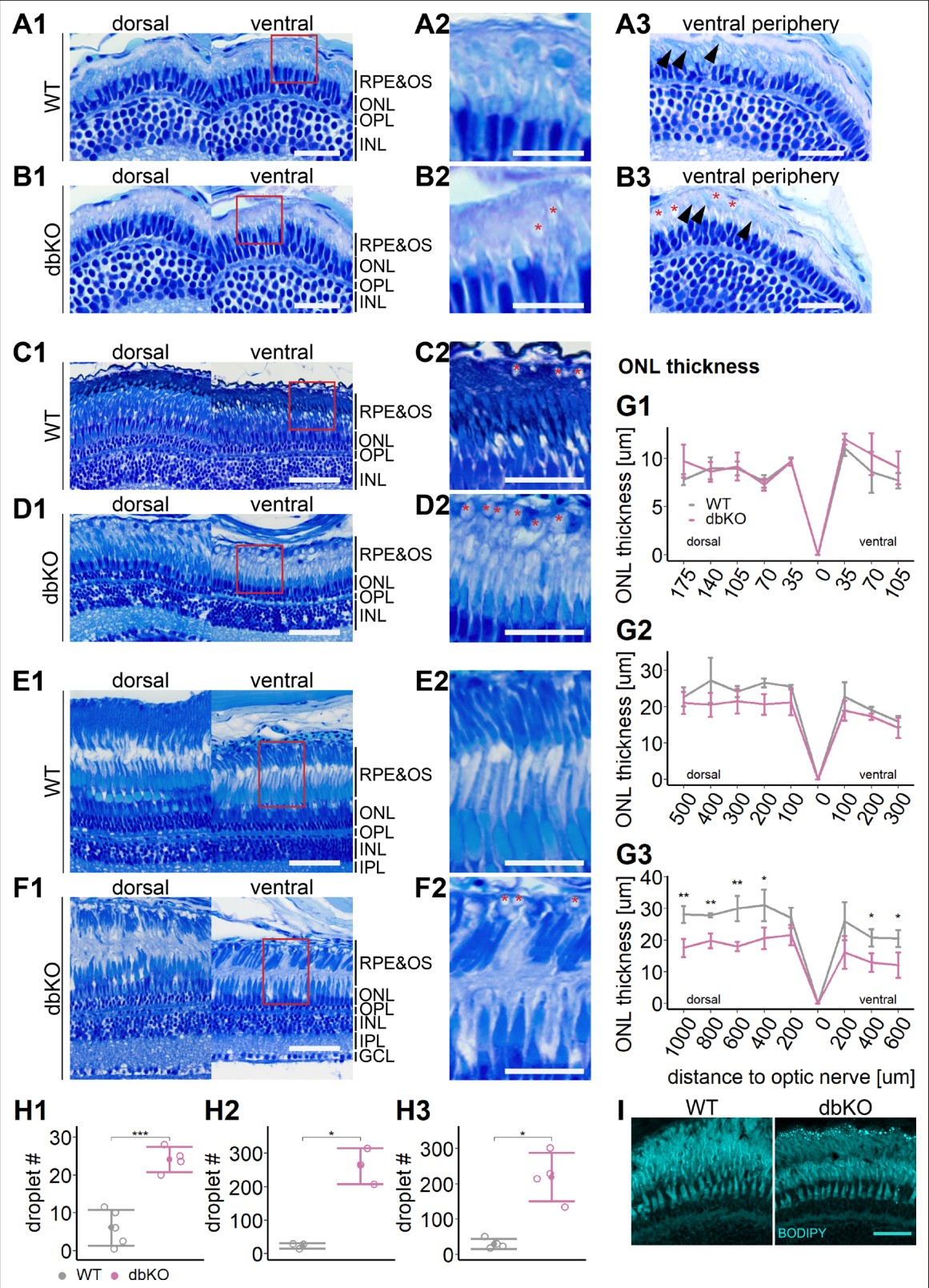

**Figure 7.** Accumulation of subretinal lipid deposits, dysmorphic outer segments, and age-dependent retinal thinning in double-knockout (KO). (A1–F1) Central retinal semithin plastic sections (dorsal and ventral of the optic nerve) of wild type (WT) and double-KO 5 days postfertilization (dpf) larvae (A, B), 1- month-old juveniles (C, D), and adults (E, F). (A2–F2) Closeup of area in red box. Subretinal lipid deposits in double-KO retinas are found already at 5 dpf and increase in number with age. Occasionally, these structures are found in wild-type controls. Black arrows: rod outer segments. Red star:

*Figure 7 continued on next page*

*Figure 7 continued*

subretinal lipid deposits. (**A3, B3**) Ventral peripheral retina. Black arrows indicate rod outer segments. These are less abundant, but not completely absent in double-KO retinas. (**G**) Quantification of outer nuclear layer (ONL) thickness in 5 dpf (**G1**), 1- month-old (**G2**), and adult (**G3**) retinas. Data are from $n \geq 3$ biological replicates per genotype and age. Shown is the mean ± SD at specified distances from the optic nerve in central retinal sections (dorsal and ventral). Effect of genotype on ONL thickness (repeated-measures analysis of variance [ANOVA]) at 5 dpf (**G1**) [$F_{(1,7)}$ = 3.701, p = 0.096], at 1 month (**G2**) [$F_{(1,4)}$ = 4.459, p = 0.102], and in adults (**G3**) [$F_{(1,6)}$ = 44.074, p < 0.001]. Pairwise comparisons of dbKO vs. WT by region were calculated with Benjamini–Hochberg correction for multiple comparisons. (**H**) Quantification of the number of lipid droplets throughout a central retinal section in 5 dpf (**H1**), 1- month-old (**H2**), and adult (**H3**) retinas. Comparison of double-KO vs. wild type was performed using t-test: (**H1**) [$t_{(6.94)}$ = −6.72, p < 0.001], (**H2**) [$t_{(2.08)}$ = −7.65, p = 0.015], and (**H3**) [$t_{(3.25)}$ = −5.39, p = 0.0102]. Individual data points (open circles) from $n \geq 3$ biological replicates per genotype and age are shown with mean ± standard deviation (SD). (**I**) Staining for neutral lipids with BODIPY TR methyl ester confirms accumulation of lipid droplets in the subretinal space of the double-KO retina. The example shows adult retinas of the respective genotype. Significance levels depicted as asterisks: *p < 0.05, **p < 0.01, ***p < 0.001. Scale bars: (**A1, A3, B1, B3**) 20 μm, (**A2, B2**) 10 μm, (**C1–F1**) 50 μm, (**C2–F2**) 25 μm, and (**I**) 10 μm. See *Figure 7—source data 1* for measurements in G and H.

The online version of this article includes the following source data for figure 7:

**Source data 1.** Source data for *Figure 7*.

the environment of foveal cones, even if rods are not completely absent. At this age, the accumulation of retinyl esters was not very pronounced yet and the number of lesions was much lower than in older retinas. On the other hand, juvenile and adult retinas of zebrafish are cone rich but contain roughly 40 % of rods that are regularly interspersed throughout the retina (*Fadool, 2003*) in contrast to the localized distribution in humans. The older zebrafish retina, therefore, resembles more the outer macular region. Complementary pathways for cones are thought to help them compete for available chromophores with the more abundant rod photoreceptors in the human retina. The density of rods peaks just outside of the macular region. This is likely to exacerbate the need for alternative and larger chromophore sources. The dramatic increase of enlarged retinosomes at the age of 1 month seems therefore in line with the hypothesis that the white puncta are associated with the presence of rods. It seems likely, that the different visual cycle pathways in RPE and MGC would be fine-tuned according to the type of cells they are in contact with. In line with this idea, single-cell transcriptomic analysis of the human retina and RPE (*Voigt et al., 2019a*; *Voigt et al., 2019b*) revealed differences in peripheral and macular gene expression. Based on these datasets (*Kiser and Palczewski, 2021*) suggested higher importance for MGC-expressed *RLBP1* in the peripheral retina compared to the fovea. The MGC-mediated cycle may become more important in later stages with the increase of rod contribution as it was previously suggested in *Fleisch et al., 2008*. This could explain the absence of an ERG phenotype in *rlbp1b−/−* larvae and the only subtle changes in the capacity of double-KO (compared to *rlbp1a−/−*) to respond to strong bleaching lights and to recover thereafter. It remains to be determined whether the knockout of *rlbp1b* will have a more pronounced effect on retinal function at later stages when rods are more abundant. But it is worthwhile to mention that Cralbpb influenced the retinyl ester storage in adult *rlbp1b−/−* and double-KO in the present study.

Differences in density and distribution of photoreceptors and the expression pattern or activity of visual cycle proteins could in part explain the discrepancies between different animal models with mutations in CRALBP. Also, the differences in 11cisRE content between species might add to this. In addition, the light environment to which the photoreceptors are exposed during life (nocturnal vs. diurnal species) may contribute to the phenotype. Together, these species-specific differences may provide an explanation for the absence of subretinal lipid droplets in mouse eyes.

In conclusion, knockout of RPE-expressed *rlbp1a* in zebrafish recapitulates the functional impairment and degenerative phenotype in *RLBP1* disease. Importantly, our model shows an increase in retinyl esters and enlarged retinosomes, providing an explanation for the yellow-white fundus lesions commonly found in *RLBP1* patients (see *Figure 9* for a graphical overview of our working model). It is therefore an excellent model not only to further dissect the disease mechanism, but also to develop potential cures for *RLBP1*-associated disease.

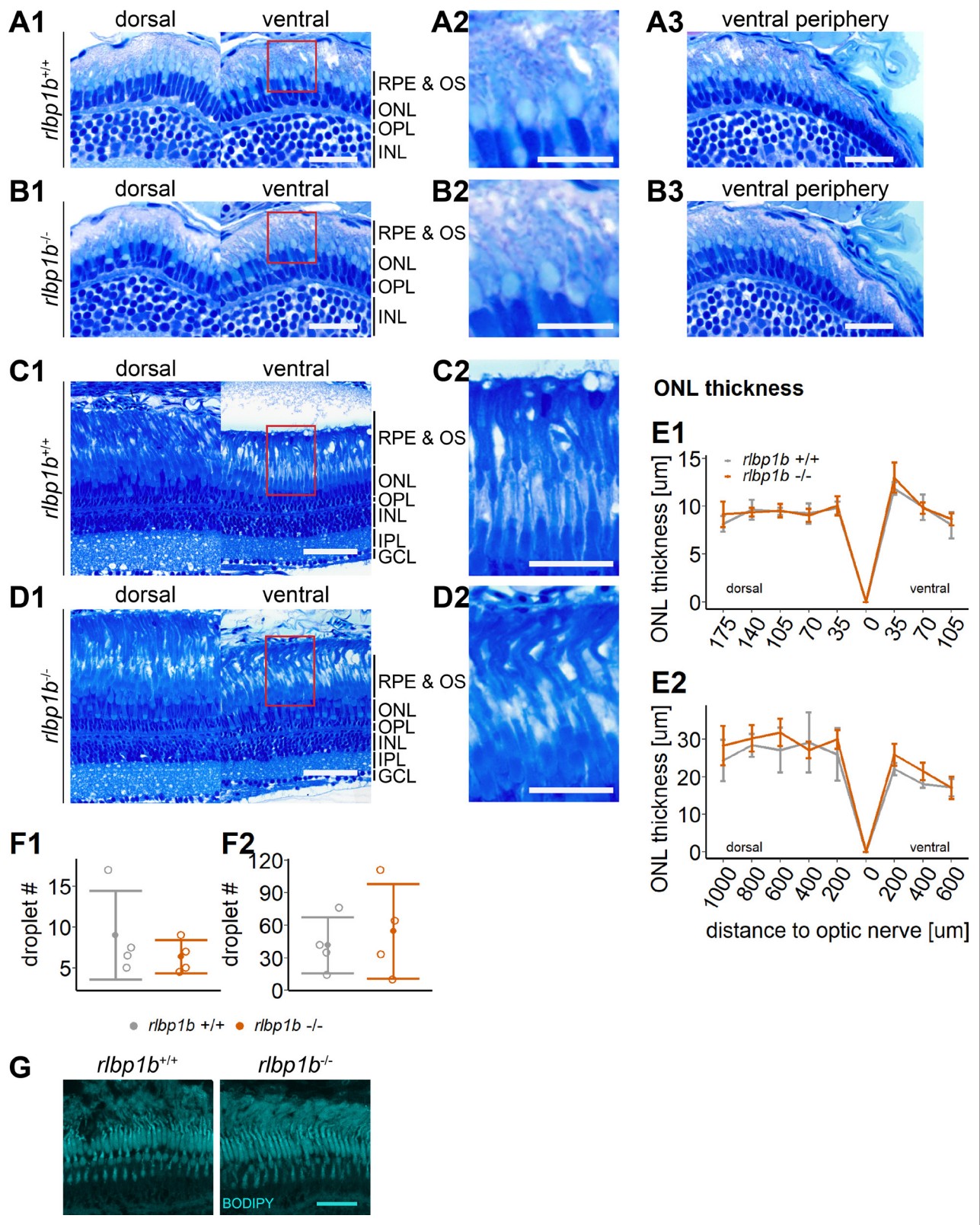

**Figure 8.** Normal retinal morphology and no accumulation of subretinal lipids in *rlbp1b*-knockout (KO). (**A1–D1**) Central retinal semithin plastic sections (dorsal and ventral of the optic nerve) of *rlbp1b*+/+ and *rlbp1b*−/− 5 days postfertilization (dpf) larvae (**A, B**), and adults (**C, D**). (**A2–D2**) Closeup of area in red box. (**A3, B3**) Ventral peripheral retina. (**E**) Quantification of outer nuclear layer (ONL) thickness in 5 dpf (**E1**) and adult (**E2**) retinas. Data are from *n* ≥ 3 biological replicates per genotype and age. Shown is the mean ± standard deviation (SD) at specified distances from the optic nerve in central

*Figure 8 continued on next page*

*Figure 8 continued*

retinal sections (dorsal and ventral). Effect of genotype on ONL thickness (repeated-measures analysis of variance [ANOVA]) at 5 dpf (**E1**) [$F_{(1,8)}$ = 0.881, p = 0.38] and in adults (**E2**) [$F_{(1,6)}$ = 1.278, p = 0.30]. (**F**) Quantification of the number of lipid droplets throughout a central retinal section in 5 dpf (**F1**) and adult (**F2**) retinas. Comparison of *rlbp1b−/−* vs. *rlbp1b+/+* was performed using *t*-test: (**F1**) [$t_{(3.84)}$ = 0.904, p = 0.42] and (**F2**) [$t_{(4.86)}$ = −0.503, p = 0.64]. Individual data points (open circles) from $n \geq 3$ biological replicates are shown per genotype and age with mean ± standard deviation (SD). (**G**) Staining for neutral lipids with BODIPY TR methyl ester reveals a healthy adult retina in both *rlbp1b+/+* and *rlbp1b−/−*. Scale bars: (**A1, A3, B1, B3**) 20 µm, (**A2, B2**) 10 µm, (**C1, D1**) 50 µm, (**C2, D2**) 25 µm, and (**G**) 10 µm. See *Figure 8—source data 1* for measurements in E and F.

The online version of this article includes the following source data for figure 8:

**Source data 1.** Source data for *Figure 8*.

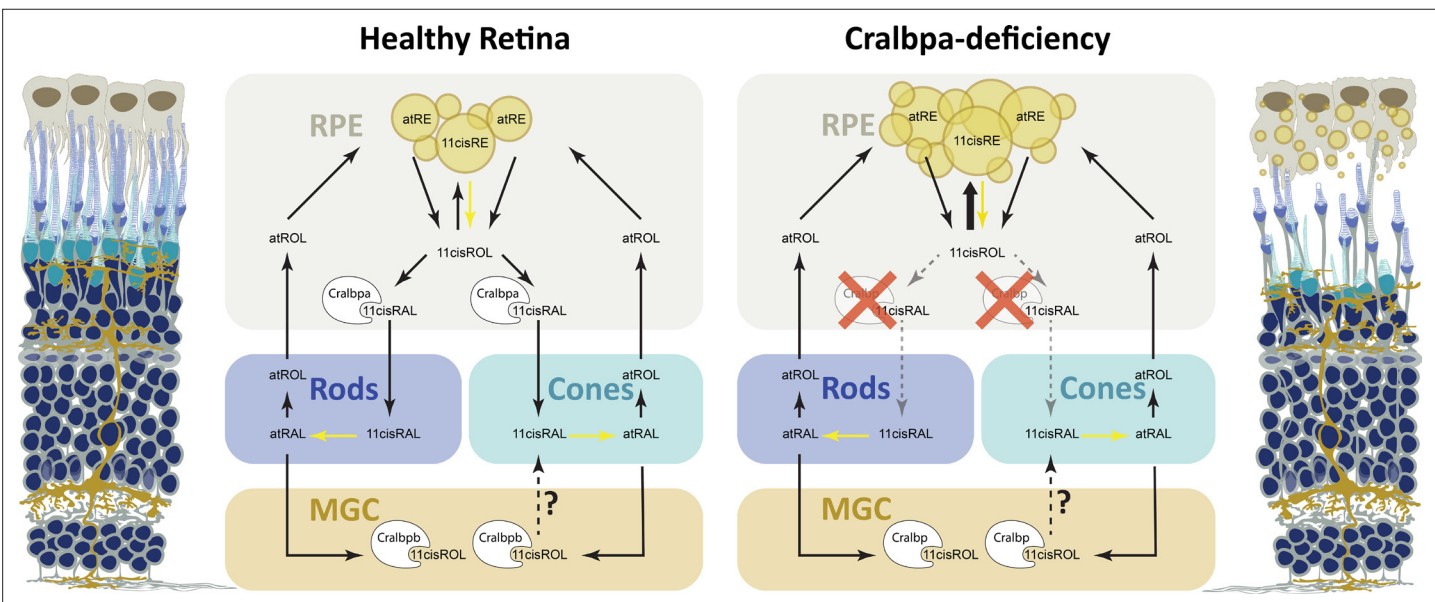

**Figure 9.** Retinoid metabolism in healthy and Cralbpa-deficient retinas. Illustration of a healthy (leftmost) and Cralbpa-deficient (rightmost) retina with the three neural layers (nuclei in dark blue). Rod and cone photoreceptors are depicted with dark blue or turquoise outer segments, respectively. Both retinal pigment epithelial (RPE) cells (gray-brown) and Müller glia (yellow-brown) contact photoreceptors and are potential sources for chromophore. Between the illustrations, schematics for retinoid metabolism in healthy and Cralbpa-deficient retinas are shown with black arrows indicating the sequence of reactions or transport/diffusion (for simplicity, enzymes or transport proteins other than Cralbp are not shown). In healthy retinas, Cralbpa facilitates the generation of 11cisRAL from both atRE (via the canonical visual cycle) and 11cisRE (through light-dependent hydrolysis, yellow arrow). Therefore, lack of RPE-expressed Cralbpa leads to a reduction of 11cisRAL (gray, dotted arrows) and an accumulation of retinyl ester (bold, black arrow), which manifests as an increased number of enlarged retinosomes in RPE cells (yellow circles in schematic and illustration on the right). Both rod and cone outer segment morphology is affected and with age the outer nuclear layer is thinning and the RPE may become atrophic. The contribution of Müller glial cell-expressed Cralbpb to chromophore regeneration is smaller than expected and the enzymes involved as well as the retinoid species supplied to photoreceptors (11cisRAL/11cisROL?) remain to be further investigated.

# Materials and methods

## Key resources table

| Reagent type (species) or resource | Designation | Source or reference | Identifiers | Additional information |
|---|---|---|---|---|
| Gene (*Danio rerio*) | *rlbp1a* | Ensembl (GRCz11) | Ensembl (GRCz11): ENSDARG00000045808 | |
| Gene (*Danio rerio*) | *rlbp1b* | Ensembl (GRCz11) | Ensembl (GRCz11): ENSDARG00000012504 | |
| Strain, strain background (*Danio rerio*) | Zebrafish: wild type Tü | KIT, European Zebrafish Resource Center (EZRC) | #1,173 | |

*Continued on next page*

*Continued*

| Reagent type (species) or resource | Designation | Source or reference | Identifiers | Additional information |
|---|---|---|---|---|
| Strain, strain background (*Danio rerio*) | Zebrafish: wild type AB | KIT, European Zebrafish Resource Center (EZRC) | #1,175 | |
| Genetic reagent (*Danio rerio*) | rlbp1a$^{-19bp}$ | This paper | mutant line | Methods: CRISPR/Cas9 mutagenesis; line available at corresponding author's lab |
| Genetic reagent (*Danio rerio*) | rlbp1b$^{-18bp}$ | This paper | mutant line | Methods: CRISPR/Cas9 mutagenesis; line available at corresponding author's lab |
| Sequence-based reagent | rlbp1a_target_oligo | This paper | CRISPR target oligo | Methods, *Table 1*; *Table 2* ordered at Microsynth |
| Sequence-based reagent | rlbp1b_target_oligo | This paper | CRISPR target oligo | Methods, *Table 1*; ordered at Microsynth |
| Sequence-based reagent | rev_oligo | This paper | CRISPR reverse oligo | Methods, *Table 1*; ordered at Microsynth |
| Sequence-based reagent | rlbp1a_s | This paper | PCR primer | Methods, *Table 1*; ordered at Microsynth |
| Sequence-based reagent | rlbp1a_as | This paper | PCR primer | Methods, *Table 1*; ordered at Microsynth |
| Sequence-based reagent | rlbp1b_s | This paper | PCR primer | Methods, *Table 1*; ordered at Microsynth |
| Sequence-based reagent | rlbp1b_as | This paper | PCR primer | Methods, *Table 1*; ordered at Microsynth |
| Sequence-based reagent | rlbp1a | This paper | sense and antisense qPCR primer | Methods, *Table 3*; ordered at Microsynth |
| Sequence-based reagent | rlbp1b | This paper | sense and antisense qPCR primer | Methods, *Table 3*; ordered at Microsynth |
| Sequence-based reagent | rho | This paper | sense and antisense qPCR primer | Methods, *Table 3*; ordered at Microsynth |
| Sequence-based reagent | actb1 | This paper | sense and antisense qPCR primer | Methods, *Table 3*; ordered at Microsynth |
| Sequence-based reagent | rpl13a | NM212784 | sense and antisense qPCR primer | Methods, *Table 3*; ordered at Microsynth |
| Sequence-based reagent | g6pd | This paper | sense and antisense qPCR primer | Methods, *Table 3*; ordered at Microsynth |
| Antibody | CRALBPas (rabbit antiserum) | gift from Dr. John Saari, University of Washington, Seattle, WA | | (1:300) |
| Antibody | glutamine synthetase GS (mouse monoclonal) | Merck | Mab302 | (1:200) |
| Antibody | RPE65 (guinea pig polyclonal) | gift from Dr. Christian Grimm, University of Zurich, Zurich, Switzerland | | (1:100) |
| Peptide, recombinant protein | Cas9-GFP | gift from Dr. Christian Mosimann | MJ922 | |
| Chemical compound, drug | CellTrace BODIPY TR methyl ester | Thermo Fisher Scientific | C34556 | (1:200) |

*Continued on next page*

*Continued*

| Reagent type (species) or resource | Designation | Source or reference | Identifiers | Additional information |
|---|---|---|---|---|
| Commercial assay or kit | Technovit 7,100 | Kulzer | 14,653 | |
| Commercial assay or kit | ReliaPrep RNA Tissue Miniprep System | Promega | Z6111 | |
| Commercial assay or kit | SuperScript III First-Strand Synthesis SuperMix | Thermo Fisher Scientific | 18080400 | |
| Commercial assay or kit | SsoAdvanced Universal SYBR Green Supermix | Bio-Rad Laboratories | 172–5270 | |
| Commercial assay or kit | MEGAshortscript T7 Transcription Kit | Thermo Fisher Scientific | AM1354 | |
| Commercial assay or kit | Megaclear Kit | Thermo Fisher Scientific | AM1908 | |
| Other | Entellan new | Merck | 107,961 | |
| Other | Tissue Freezing Medium clear | Electron Microscopy Sciences, Hatfield, PA | 72,593 | |
| Software, algorithm | Fiji ImageJ | *Schindelin et al., 2012* | | |
| Software, algorithm | R base | *R Development Core Team, 2020*. R: A language and environment for statistical computing. R Foundation for Statistical Computing, Vienna, Austria. URL https://www.R-project.org/. | | |
| Software, algorithm | rstatix | *Kassambara, 2020*. rstatix: Pipe-Friendly Framework for Basic Statistical Tests. R package version 0.6.0. https://CRAN.R-project.org/package=rstatix | R package | |
| Software, algorithm | ggplot2 | *Wickham, 2016*. | R package | |
| Software, algorithm | ggpubr | *Kassambara, 2020*. ggpubr: 'ggplot2' Based Publication Ready Plots. R package version 0.4.0. https://CRAN.R-project.org/package=ggpubr | R package | |
| Software, algorithm | Adobe InDesign | Adobe | | |
| Software, algorithm | Adobe Illustrator | Adobe | | |

## Zebrafish maintenance

Zebrafish (*Danio rerio*) were kept at 26 °C under a 14 hr/10 h light/dark cycle and bred by natural spawning. Embryos were raised at 28 °C in E3 embryo medium (5 mM NaCl, 0.17 mM KCl, 0.33 mM $CaCl_2$, 0.33 mM $MgSO_4$) and staged according to development in days post fertilization (dpf). The wild types used for CRISPR/Cas9 mutagenesis were of the Tü (for the *rlbp1a* line) and AB (for the *rlbp1b* line) strain or Tü:AB hybrid (double-KO resulting from a cross between the *rlbp1a* and *rlbp1b* lines). All zebrafish protocols complied with internationally recognized guidelines for the use of zebrafish in biomedical research and experiments were approved by local authorities (Kantonales Veterinäramt TV4206).

## CRISPR/Cas9 mutagenesis

Generation of stable homozygous KO lines for *rlbp1a* and *rlbp1b* was achieved by using the CRISPR/Cas9 approach. Oligos for single guide RNA (sgRNA) synthesis were designed using the CHOPCHOP web application (https://chopchop.cbu.uib.no/): 5'-GGAGCTCAGAGGAATAATCA-3' and 5'-GGCAG-GCCAAGGAGATGAGG-3' were chosen for *rlbp1a* and *rlbp1b*, respectively. Synthesis of sgRNAs was achieved by polymerase chain reaction (PCR) using a high fidelity Phusion polymerase (New England Bio Labs., Ipswich, MA) and in vitro transcription of the product: Target-specific oligos (target_oligo) were synthesized (Microsynth AG, Balgach, Switzerland) containing a T7 promoter site, the above target-specific sequence, and part of the tracerRNA sequence. A second, partially overlapping, oligo (rev_oligo) was synthesized (Microsynth AG, Balgach, Switzerland) to contain the reverse tracerRNA

**Table 1.** Target oligos (underlined: gene-specific sequence; bold: tracerRNA overlap) for single guide RNA (sgRNA) synthesis and genotyping primers.

| Name | Sequence 5'–3' | Length | Exon |
|------|----------------|--------|------|
| rlbp1a_target_oligo | GAAATTAATACGACTCACTATAGGGAGCTCAGAGGAATAATCA**GTTTTAGAGCTAGAAATAGC** | 62 bp | 4 |
| rlbp1b_target_oligo | GAAATTAATACGACTCACTATAGGCAGGCCAAGGAGATGAGG**GTTTTAGAGCTAGAAATAGC** | 62 bp | 4 |
| rev_oligo | AAAGCACCGACTCGGTGCCACTTTTTCAAGTTGATAACGGACTAGCCTTATTTTAACTT**GCTATTTCTAGCTCTAAAAC** | 79 bp | – |
| rlbp1a_s | GTTAGCATGCTTCATTGAGG | 20 bp | 4 |
| rlbp1a_as | CGGATGAACCTCACAAGC | 18 bp | 4 |
| rlbp1b_s | TGAACGAAACAGATGAGAAGAG | 22 bp | 4 |
| rlbp1b_as | TCTGGCCACGTCAAACTTG | 19 bp | 4 |

sequence. After PCR and cleanup, the dsDNA was in vitro transcribed (MEGAshortscript T7 Transcription Kit, Ambion, Thermo Fisher Scientific, Waltham, MA) and the resulting sgRNA was purified (Megaclear Kit, Ambion, Thermo Fisher Scientific, Waltham, MA) and precipitated using ethanol.

Injection mixes containing 400 ng/µl sgRNA, 1000 ng/µl Cas9-GFP (MJ922; kind gift of C. Mosimann), and 300 mM KCl were incubated for 10 min at 37 °C to allow for ribonucleoprotein complex formation. One-cell stage embryos were injected with 1 nl of injection mix and raised to adulthood. Injected F0 fish were outcrossed with wild types and the offspring were genotyped using primers designed around the target region (see *Table 1*) to check for successful germline transmission. F1 animals carrying the same mutation (−19 bp in *rlbp1a* and −18 bp in *rlbp1b*, see *Table 2*) were incrossed to obtain homozygous mutants (F2).

## Immunohistochemistry

Whole larvae or adult eyes were fixed in 4 % formaldehyde for 30 min at room temperature (RT). After washing with 1× phosphate-buffered saline (PBS), the tissue was cryo-protected in 30 % sucrose overnight at 4 °C. Samples were embedded in Tissue Freezing Medium clear (EMS-72593, Electron Microscopy Sciences, Hatfield, PA). Samples were sectioned at 12 µm thickness using a Cryostar NX50 (Thermo Fisher Scientific, Waltham, MA) and collected on SuperFrost Plus Adhesion Slides (Thermo Fisher Scientific, Waltham, MA). Sections were spread across multiple slides and wild-type controls were collected next to heterozygous and/or homozygous samples on the same slide to achieve equal conditions during staining and washing steps for all genotypes. Sections were stored at least once overnight at −80 °C. Before staining, the slides were dried for 30 min at 37 °C. Sections were bleached in 10 % $H_2O_2$ solution in PBS for 30 min at 65 °C. Slides were washed with 1× PBS and mounted on a Shandon Sequenza Slide Rack (Thermo Fisher Scientific, Waltham, MA). Slides were incubated with blocking solution (10 % normal goat serum, 1 % BSA in 0.3 % PBS Triton X-100) for 2 hr at RT and with primary antibodies overnight at 4 °C. Samples were washed with 1× PBS and secondary antibodies were applied for 1.5 hr at RT in darkness. After washing with 1× PBS, samples were stained with CellTrace BODIPY TR methyl ester (1:200 in PDT [1 % dimethyl sulfoxide, 0.1 % Triton X-100 in PBS], cat# C34556, Thermo Fisher Scientific, Waltham, MA) for 20 min at RT and

**Table 2.** Wild-type sequences of rlbp1a and rlbp1b in the target region and the corresponding mutated sequence (underlined: target sequence; dashes: deletion).

**Mutation sequences 5'–3':**

| | |
|---|---|
| rlbp1a | CGTCAGCTGTGAAGGAGCTCAGAGGAATAATCAAGGAGAAGGCAGAGACT |
| rlbp1a −19 bp | CGTCAGCTGTG------------------TCAAGGAGAAGGCAGAGACT |
| rlbp1b | CCATGATTAAAGACAAGGCAGGCCAAGGAGATGAGGTGGCCAAAACTGTG |
| rlbp1b −18 bp | CCATGATTAAAGACA-----------------AGGTGGCCAAAACTGTG |

**Table 3.** primers for real-time polymerase chain reaction (PCR).

| Gene | Sense 5′–3′ | Antisense 5′–3′ |
| --- | --- | --- |
| *rlbp1a* | GTCAAACCCCTGATGAAGAGC | GTGATCTTGCCGTCATATTTGG |
| *rlbp1b* | CTGCGTGCCTACTGTGTAATCC | ATCACATGCACAGCCTTAAACC |
| *rho* | GGTCGCTTGTAGTACTGGC | ATGTAACGCGACCAGCC |
| *actb1* | CAGACATCAGGGAGTGATGGTTGG | CAGATCTTCTCCATGTCATCCCAG |
| *rpl13a* | TCTGGAGGACTGTAAGAGGTATGC | AGACGCACAATCTTGAGAGCAG |
| *g6pd* | CTGGACCTGACCTACCATAGCAG | AGGCTTCCCTCAACTCATCACTG |

washed again with 1× PBS. Finally, samples were covered with Mowiol (Polysciences, Warrington, PA, USA) containing DABCO (1,4-diazo-cyclo[2.2.2] octane, Sigma-Aldrich, Steinheim, Germany) and imaged with a TCS LSI confocal microscope (Leica Microsystems, Heerbrugg, Switzerland) using ACS APO 40.0 × 1.15 OIL objective (11507901). Rabbit CRALBP antiserum (1:300, kindly provided by Dr. John Saari, University of Washington, Seattle, WA) was used to detect Cralbpa and Cralbpb. Mouse anti-glutamine synthetase (GS, 1:200, Mab302, Merck KGaA, Darmstadt, Germany) was used to stain Müller glia cells and guinea pig anti-RPE65 (1:100, kindly provided by Dr. Christian Grimm, University of Zurich, Zurich, Switzerland) for RPE cells. Secondary antibodies were goat anti-guinea pig, anti-rabbit, and anti-mouse IgG conjugated to Alexa 488, 568, or 647 (1:500, Molecular Probes, Thermo Fisher Scientific, Waltham, MA). Images were adjusted for brightness and contrast using ImageJ Fiji software (*Schindelin et al., 2012*).

## Real-time PCR

Both eyes of *n* = 50 larvae per sample or one adult eye per sample were collected and in total three biological replicates were used for each genotype. RNA was isolated using the ReliaPrep RNA Tissue Miniprep System (Promega, Madison, WI). SuperScript III First-Strand Synthesis SuperMix (Thermo Fisher Scientific, Waltham, MA) with a mix of oligo dT primers and random hexamers was used for reverse transcription of the RNAs. Primers used for real-time PCR are summarized in *Table 3*. Real-time PCR was performed using SsoAdvanced Universal SYBR Green Supermix (catalog #172-5270, Bio-Rad Laboratories, Hercules, CA) on a Bio-Rad CFX96 C1000 Touch Thermal Cycler (Bio-Rad Laboratories, Hercules, CA) using 1 ng template cDNA. Relative normalized expression was calculated by normalizing to the wild-type samples and using *actb1*, *rpl13a*, and *g6pd* as reference genes.

## Semithin sections

Whole larvae and adult eyes were fixed at least overnight in 4 % formaldehyde at 4 °C and embedded in Technovit 7,100 (Kulzer, Wehrheim, Germany). Using a microtome (LeicaRM2145, Leica Microsystems, Nussloch, Germany) samples were sectioned at 3 µm thickness and stained in Richardson's solution (1 % methylene blue, 1 % borax) for 5 s, washed for 10 min in ddH$_2$O, and coverslipped with Entellan mounting medium (Merck, Darmstadt, Germany). Images were acquired on an Olympus BX61 microscope (Olympus, Shinjuku, Tokyo, Japan) and adjusted for brightness and contrast using Fiji ImageJ (*Schindelin et al., 2012*). Quantifications were performed on whole retinal sections using Fiji ImageJ. Filenames were randomized to mask the genotype during quantification. ONL thickness was measured dorsally and ventrally every 35 µm (5 dpf), every 100 µm (1 month), and every 200 µm (adults), starting at the optic nerve. Lipid droplets were counted manually with the Cell Counter plugin.

## Samples for retinoid analysis and light treatments

Adult zebrafish eyes were collected in ambient room light and immediately frozen in liquid nitrogen and stored at −80 °C before HPLC analysis. For retinoid measurements in zebrafish larvae, light treatments were performed similar to previously described methods (*Babino et al., 2015*). Samples were grouped according to light condition: overnight DA, DA and exposure to ambient light (ambient), DA and 30 min bleach (30 min BL), DA and 1 hr bleach (1 hr BL), and DA, 1 hr bleach followed by 1 hr redark adaptation (1 hr reDA). Zebrafish larvae (*n* = 100 per group) were sequentially placed in darkness overnight and immediately collected for HPLC analysis or subjected to indicated light treatments.

For light treatments, larvae were placed into open 6 cm Petri dishes wrapped with aluminum foil and exposed to either ambient room light or a strong bleaching light with 20'000 lux using a KL 1500 HAL halogen cold light source (Schott, Mainz, Germany) for the indicated time (30 min or 1 hr). Data are the results of three independent experiments for each age group and KO line.

## Electroretinography

Standard white-light electroretinography (ERG) was performed as described (**Makhankov et al., 2004**), with adaptations. For DA ERG, 5 dpf larvae were placed in darkness for at least 1 hr before measurements. Larvae were sacrificed before the removal of one eye for measurements. For ERG on single mutants, offspring of heterozygous crosses were used and the trunks were used for geno-typing after measurements. Five stimulations of 100 ms duration and with increasing light intensity (100 % corresponding to 24'000 µW/cm$^2$) were applied with an interstimulus interval of 15'000 ms. For light-adapted ERGs, an additional background illumination of 20 µW/cm$^2$ was applied and larvae were not DA before measurements. For bleaching experiments, larvae were exposed to BL from a KL 1500 HAL halogen cold light source (as described in light treatments for retinoid analysis) for 1 hr before measurements. To reduce the number of larvae as well as resources for genotyping, we did not directly use double-heterozygous incrosses for double-KO ERGs. Instead, we incrossed wild-type siblings from such a cross to yield closely related offspring as control and double-KO fish were incrossed to yield homozygous mutant offspring. Measurements were blinded by allocating wild type and double-KO larvae to wells of a 24-well plate according to a random key. For experiments including different light conditions, larvae were derived from the same clutch for all light conditions, but measurements were derived from individual retinas, due to the invasive nature of our protocol.

## HPLC analysis of retinoids

For retinoid extraction, a whole eye from wild type and mutant adult zebrafish or 100 zebrafish mutant larvae were transferred to 2 ml Eppendorf microcentrifuge tubes (Eppendorf, Hamburg, Germany). 200 µl of 2 M hydroxylamine (pH 8) and 200 µl of methanol were added and eyes or larvae were homogenized and incubated for 10 min at RT. Then, 400 µl of acetone was added to the homogenate and the mixture was vortexed. Retinoids were then extracted with 500 µl of hexanes by vortexing vigorously. For phase separation, the mixture was centrifuged at 5000 × *g* for 1 min at RT. The upper organic phase was collected and transferred into a fresh tube. This was repeated and the collected organic phases containing the retinoids were dried using the SpeedVac system (Vacufuge Plus, Eppen-dorf, Hamburg, Germany). The dried samples were reconstituted in hexanes and used for HPLC sepa-ration. Retinoid analyses were performed using normal phase Zorbax Sil (5 mm, 4.6 3 150 mm) column (Agilent Technologies, Santa Clara, CA, USA). The retinoids were eluted by stepwise gradient starting with 0.5 % ethyl acetate in hexane over 15 min, followed by 20 min of 10 % ethyl acetate in hexane with a continuous flow rate of 1.4 ml/min. Finally, the column was equilibrated for 10 min with 0.5 % ethyl acetate in hexane. For molar quantification of retinoids, the HPLC was scaled with authentic standard compounds purchased by Toronto Chemicals (Toronto, Canada).

## Statistical analysis

The number of biological replicates is defined in the figure legends as '*n*'. It describes the number of individual larvae/fish used in the study. In **Figure 2B and D**, it refers to samples consisting of a pool of 100 larvae, each. Statistical analysis was performed as indicated in the figure legends using R base (**R Development Core Team, 2020**. R: A language and environment for statistical computing. R Foun-dation for Statistical Computing, Vienna, Austria. URL https://www.R-project.org/.) and the rstatix package (**Kassambara, 2020**. rstatix: Pipe-Friendly Framework for Basic Statistical Tests. R package version 0.6.0. https://CRAN.R-project.org/package=rstatix). Data were plotted using ggplot2 (H. Wickham. ggplot2: Elegant **Wickham, 2016**.) and ggpubr (**Kassambara, 2020**. ggpubr: 'ggplot2' Based Publication Ready Plots. R package version 0.4.0. https://CRAN.R-project.org/package=ggpubr) packages.

Final figures were assembled using Adobe InDesign and the illustration in **Figure 9** was created in Adobe Illustrator.

## Acknowledgements

We thank John Saari for providing the CRALBP antiserum and Christian Grimm for the RPE65 antibody. We thank Martin Walther and Hannah Kämper for help with genotyping and excellent fish care. The latter was supported by Heidi Möckel and Kara Kristiansen. This work was supported by Robert und Rosa Pulfer-Stiftung and SNF 31,003 A_173083. The von Lintig laboratory is supported by grants from the National Eye Institute (EY028121 and EY020551).

## Additional information

### Funding

| Funder | Grant reference number | Author |
| --- | --- | --- |
| Schweizerische Nationalfonds | 31003A_173083 | Stephan CF Neuhauss |
| National Eye Institute | EY028121 | Johannes von Lintig |
| National Eye Institute | EY020551 | Johannes von Lintig |
| Robert und Rosa Pulfer Stiftung | | Domino K Schlegel |

The funders had no role in study design, data collection and interpretation, or the decision to submit the work for publication.

### Author contributions

Domino K Schlegel, Conceptualization, Data curation, Formal analysis, Investigation, Methodology, Project administration, Visualization, Writing - original draft, Writing - review and editing; Srinivasagan Ramkumar, Formal analysis, Investigation, Methodology, Visualization, Writing - review and editing; Johannes von Lintig, Conceptualization, Funding acquisition, Investigation, Methodology, Project administration, Resources, Supervision, Writing - review and editing; Stephan CF Neuhauss, Conceptualization, Funding acquisition, Methodology, Project administration, Resources, Supervision, Writing - review and editing

### Author ORCIDs

Domino K Schlegel  http://orcid.org/0000-0001-6194-4695
Stephan CF Neuhauss  http://orcid.org/0000-0002-9615-480X

### Ethics

Holding and experimental permits have been granted by the Zurich cantonal veterinary office (TV4206).

### Decision letter and Author response

Decision letter https://doi.org/10.7554/eLife.71473.sa1
Author response https://doi.org/10.7554/eLife.71473.sa2

## Additional files

### Supplementary files

• Transparent reporting form

### Data availability

All data generated or analysed during this study are included in the manuscript. Source data files for all figures have been provided.

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
