## [Decision Letter]

**Acceptance summary:**

In mammals the cellular retinaldehyde binding protein, CRALBP, is expressed in the pigment epithelium (RPE) and in Müller glial cells in the retina. Zebrafish has two copies of the gene, each expressed in one of the cell types. By knocking out each gene with CRISPR/Cas9, the authors could show that it is the copy expressed in the RPE that is essential for turnover of retinal and for cone function. Thus, the zebrafish gene duplication suggests that the RPE role of CRALBP is the important one also in humans, implying the RPE as target for future gene therapy in humans with mutations in CRALBP.

**Decision letter after peer review:**

Thank you for submitting your article "Disturbed retinoid metabolism upon loss of rlbp1a impairs cone function and leads to subretinal lipid deposits and photoreceptor degeneration in the zebrafish retina" for consideration by *eLife*. Your article has been reviewed by 3 peer reviewers, and the evaluation has been overseen by a Reviewing Editor and Didier Stainier as the Senior Editor. The following individuals involved in review of your submission have agreed to reveal their identity: Xesus Abalo (Reviewer #1); Gennadiy Moiseyev (Reviewer #2).

All three reviewers and the reviewing editor are impressed and very positive. Here are some of the general comments from the reviewers: The experiments have been well-designed and for the most part well-executed. The results and the conclusions extracted from them are totally sound. The manuscript is written clearly and carefully, the discussion is well-structured and touches each single point. A major strength is the continuity of the reliable, rigorous and meticulous work. The graphical abstract is beautiful in many ways; stylish, informative, complete.

The reviewers and I have a number of comments and suggestions, all of which can probably be taken care of by adding clarifications and comments in the text. Some in silico work can probably be done rather quickly and easily if the information is not yet available, especially regarding the presumed ohnology of the two genes and the possible consequences of amino acid differences between them.

I apologize that the comments may not quite follow the chronology of the manuscript's text (this happens when comments from three reviewers are merged).

Essential revisions

1) Is it clear that rlbp1a and b resulted from the teleost WGD ('ohnologs') and thus had identical sequences also in regulatory regions before starting to diverge from one another after the gene duplication? Synteny and paralogon investigations should resolve this if it hasn't been done already.

2) Lines 92-93: What is the degree of protein sequence identity between the two paralogs in zebrafish? Would the differences be likely to affect substrate binding affinity and interactions with other proteins? In silico modelling might give clues by incorporating information from human mutations.

3) Please provide additional explanation about the measurement and interpretation of 11-cis retinaldehyde (11cisRAL) levels. The text refers to "11cisRAL" as the aldehyde that binds to CRALBP. However, it is unclear whether the measurements of "11cisRAL" refer to the total of 11-cis isomer covalently bound as visual pigment (in cones and rods) in addition to the non-covalently bound 11cisRAL. It will be important for the paper to make absolutely clear what was measured, and to interpret the measurements appropriately, according to whether or not covalently bound retinaldehyde was included. If the measurements did include rhodopsin and cone opsins, then a term distinct from "11cisRAL" should be employed throughout the paper.

4) After mutagenesis by CRISP-Cas9, could first 3 exons of the rlbp1a-/- or rlbp1b-/- still be translated into a semi-functional peptide? How could you assess that?

5) Lines 134 and 608: Some justification is needed to explain why the eyes were collected "in ambient light". Were any controls done to determine whether the retinoid levels differed for eyes collected under dark-adapted conditions?

6) Lines 154-156: The authors need to clarify their reasoning for deducing "that the esters were hydrolyzed upon light exposure but not utilized". Given that Figure 2B (bottom) shows that atRE levels had increased by at least as much, is it not instead likely that retinoid recycled from the retina has been re-routed to atRE rather than to 11cisRE? How can you distinguish ester hydrolysis from reduced ester synthesis? The same considerations apply to the statement on lines 428-9 in the Discussion.

7) In the sentence "rlbp1a was essential for cone photoreceptor function and chromophore metabolism", does this mean that the rlbp1a-KO does not affect rods and scotopic vision?

8) In Figure 1C, can one say that the rlbp1b-/- is really a KO? Its expression is four orders of magnitude higher than 1a in the 1a k/o. The p-value is very high and non-significant.

9) In Figure 2, there is considerable variability in the measurements of retinoid content in WT adult animals. Between panels A, C and E, the mean WT levels differ markedly: for 11cisRAL (312, 177 and 280); for 11cisRE (12.7, 86.1 and 7.2); and for atRE (168, 460 and 187). These differences would benefit from explanation. Are these WT values significantly different from each other? Would it be better to use the WT grand means in the tests of significance for the knockouts?

10) How critical is the role of Cralbpb in the MGC? The authors suggest that these cells could serve as a storage of 11-cis-retinoids. Could the authors elaborate a little this suggestion and how it might be explored? Along a similar line, which studies would the authors consider to investigate if 11cisRAL can be taken up by cones from the MGC (the question mark in the Graphical abstract)? In addition, if the MGC transform 11cisROL in 11cisRAL but the second is not transported back to the photoreceptors, what do the MGCs do with it? A storage for bright light conditions? It would be interesting to see the authors discuss this.

11) The morphology in Figure 5 (and the consecutive ones) suggests a stronger effect of the rlbp1a-KO in the ventral retina as compared to the dorsal. If this is the case, why could it be, and which implications could it have?

12) The first sentence of the discussion says "Our results showed that cones in the zebrafish retina rely on RPE-expressed Cralbpa for chromophore regeneration and retinal function" and summarizes the results, but as this has been extensively suggested, it would perhaps be better to focus more on the lack of effect of Cralbpb in MCG on chromophore regeneration. An important novelty is the use of a subfunctionalized teleost model to study RLBP1-KO related diseases.

13) If cralbp1a is completely KO but there is still some chromophore regeneration, what explanation for this can the authors suggest?

14) What do the authors speculate that could happen with the Retinomotor circadian movements? Would the Cralbp1a-KO fish have a similar response as for the dark-adapted retina? Could these movements be altered, therefore altering the circadian chromophore mobilisation and pigment regeneration and contributing to the accumulation of the small oil droplets/retinosomes in the RPE?

15) For the insertion of the sgRNA and Cas-GFP, the authors used the injection methods. Did they try or consider using electroporation, working very well for isolated mammalian cells?

16) In Figure 1A, the CRALBPas images are rather faint, probably due to poor contrast between background autofluorescence and the real signal, especially the two dbKO images. Presumably a negative control photo exists that could be added to help distinguish between the autofluorescence (due to the fixative, for example) and the real signal.

17) Figure 3: These 5 dpf b-wave signals seem huge, at ~10 mV. For comparison, Bilotta et al., (2001) reported ~50 µV for the b-wave in 8 dpf animals, so your values are 200x higher. If the scale was meant to show "µV" then the signals would be around 5x smaller than theirs. Ideally, the reader should have access to some samples of b-wave records (rather than just measured amplitudes), in order to judge the quality of the recordings, so please provide this. As a minimum, a panel of dark-adapted ERG traces should be presented as a Supplementary Figure.

18) Before the authors can state that "We observed no pronounced changes of cone responses in Cralbpb-deficient larval eyes" they need to illustrate rlbp1b KO responses in Figure 4. At present, the double KO plots in panel B2 ("BL", and "BL+reDA") appear to show considerably greater attenuation than for the rlbp1a KO plots in panel A2, strongly suggesting an effect of Cralbpb. To make the above statement, it is necessary to present the data.

19) The authors used CRALBP antiserum to detect both Cralbpa and Cralbpb. Presumably it is difficult to generate specific antibodies if the two proteins are highly identical. Instead ISH with riboprobes specific against each paralog could be used. If the coding regions are too similar,the 3´UTR could be used. The authors have surely considered these options so a comment would be appropriate.

20) The option of generating a dKO Cralbp1a/RPE65 would be an interesting next step continuing with this research line to investigate correlation between the RPE65 genes (the are three paralogs) and rlpb1a. This could be interesting to see a key protein not in transporting but to convert all-trans retinal back to 11-cis retinal, widening the spectrum of diseases to be analysed.

21) Discussion mentions (line 495 onwards) studies that use single-cell transcriptomics. The authors are encouraged to embrace this option in the near future. It would be interesting to see the transcriptome also of the KOs with spatial resolution, and using ISS or SCRINSHOT spatial mapping for specific transcripts in intact sections in situ.

---

## [Author Response]

Essential revisions1) Is it clear that rlbp1a and b resulted from the teleost WGD ('ohnologs') and thus had identical sequences also in regulatory regions before starting to diverge from one another after the gene duplication? Synteny and paralogon investigations should resolve this if it hasn't been done already.

We performed an initial synteny analysis. The two zebrafish paralogues are located on chromosomes 7 and 25 while the human ortholog maps to chromosome 15. Both zebrafish genes are localized adjacent to *ABHD2* gene paralogs. Agreeably, the human *RLBP1* gene is flanked by an *ABHD2* gene as well. In fact, it will be interesting to determine regulatory sequences that confer RPE and MG specific expression the two zebrafish *cralbp* genes. Such analyses are within the scope of our future studies.

2) Lines 92-93: What is the degree of protein sequence identity between the two paralogs in zebrafish? Would the differences be likely to affect substrate binding affinity and interactions with other proteins? In silico modelling might give clues by incorporating information from human mutations.

Protein sequence identity between Cralbpa and Cralbpb is ~80% (see Author response image 1). Based on structural predictions the overall fold of the proteins is conserved. In humans, two mutations are described that affect retinoid binding of CRALBP: M226K abolishes retinoid binding, whereas R234W tightens the binding of the lipid. The mutations are located within the CRAL-TRIO domain that is highly conserved in the two zebrafish paralogs. However, form *in silico* models we cannot predict the affinity of the two CRALBP isoforms for 11-cis-retinol/al. Thus, we do not want to exclude that subtle differences between the paralogs modulate the binding affinity for the retinoid or the interaction of the paralogs with other proteins including Rpe65 and Rdh5. These visual cycle proteins have been previously suggested to interact with CRALBP in the RPE and are not expressed in Müller glia cells. Future studies are warranted to unravel such cell-specific protein-protein interactions.

**Author response image 1. sa2fig1:** 

3) Please provide additional explanation about the measurement and interpretation of 11-cis retinaldehyde (11cisRAL) levels. The text refers to "11cisRAL" as the aldehyde that binds to CRALBP. However, it is unclear whether the measurements of "11cisRAL" refer to the total of 11-cis isomer covalently bound as visual pigment (in cones and rods) in addition to the non-covalently bound 11cisRAL. It will be important for the paper to make absolutely clear what was measured, and to interpret the measurements appropriately, according to whether or not covalently bound retinaldehyde was included. If the measurements did include rhodopsin and cone opsins, then a term distinct from "11cisRAL" should be employed throughout the paper.

We agree with the reviewers that our terminology is misleading. In our experiments we cannot distinguish between opsin bound, CRALBP bound and free 11-cis-retinal since we treated ocular tissues with hydroxylamine prior to extraction. Thus, we changed the text as follows, see lines 115-116 of the revised manuscript:

“The term 11cisRAL always refers to protein-bound (e.g. by opsin or Cralbp) and free 11cisRAL in our experiments.”

4) After mutagenesis by CRISP-Cas9, could first 3 exons of the rlbp1a-/- or rlbp1b-/- still be translated into a semi-functional peptide? How could you assess that?

In both lines, the introduced mutations leave the initial exons unaffected. The initial exons encode a coiled coil domain with non-defined function. However, the CRAL-TRIO domain that confers the lipid/retinoid binding properties of the proteins follows downstream of the target site and putative mutant variants lack this domain. Additionally, the mutation in the *rlbp1a*-KO line leads to an early stop codon in the putative mRNA transcript. Our immunostainings showed that no protein or only very little protein is produced from the mutant gene loci, indicating that putative mutant protein variants are rapidly degraded. Together, these features of our targeting strategy strongly indicate that we are dealing with functional null alleles for both CRALBP genes.

5) Lines 134 and 608: Some justification is needed to explain why the eyes were collected "in ambient light". Were any controls done to determine whether the retinoid levels differed for eyes collected under dark-adapted conditions?

We collected the adult zebrafish eyes in ambient light to measure each intermediate of the visual cycle. We would expect that 11-cis-retinal and retinyl esters are the predominant intermediates in dark-adapted eyes. Additionally, our fish room setup makes it difficult to dark-adapt adult fish for a prolonged period of time. For the investigation of the consequences of different light conditions, we used larval fish which can be easily maintained in petri dishes that can be transferred to the dark room.

6) Lines 154-156: The authors need to clarify their reasoning for deducing "that the esters were hydrolyzed upon light exposure but not utilized". Given that Figure 2B (bottom) shows that atRE levels had increased by at least as much, is it not instead likely that retinoid recycled from the retina has been re-routed to atRE rather than to 11cisRE? How can you distinguish ester hydrolysis from reduced ester synthesis? The same considerations apply to the statement on lines 428-9 in the Discussion.

We perfectly agree with this concern of the reviewers. Our statement is an assumption and not an experimentally proven fact. Therefore, we rephrased the paragraphs and now state in lines 141-143 of the revised manuscript that:

“Based on this observation and the reduced chromophore concentration in rlbp1a-/- larvae we assume that the esters were hydrolyzed upon light exposure but not utilized for 11cisRAL production”.

And in lines 282-283:

“This result raises the possibility that 11cisRE hydrolysis can occur Cralbpa-independently.”

Moreover, we indicate the other possibility that these esters stem from retinol that was released from bleached photoreceptors.

7) In the sentence "rlbp1a was essential for cone photoreceptor function and chromophore metabolism", does this mean that the rlbp1a-KO does not affect rods and scotopic vision?

In fact, we have no reasons to assume that rods are unaffected by the mutation. However, we only tested *cone* photoreceptor function in our study. Regarding retinoid metabolism, we cannot differentiate between rods and cones with our biochemical measurements since we used whole larval or whole eye (adult) samples.

Morphologically, we observed that rods are also affected, possibly from an early age on (see e.g. Figure 5A3 vs. 5B3). This is further supported by the low levels of *rhodopsin* gene expression both in larvae and adults of the *rlbp1a*-KO and double-KO lines (see Figure 8). In the abstract we already state that “these fish developed retinal thinning and cone and rod photoreceptor dystrophy”.

Additionally we now emphasize the rod defect also in the sentence line 255 of the revised manuscript:

“We found that the knockout of rlbp1a impaired cone-driven electroretinogram (ERG) responses and led to disturbed outer segment morphology *of both rods and cones* and age-dependent thinning of the photoreceptor nuclear layer.”

8) In Figure 1C, can one say that the rlbp1b-/- is really a KO? Its expression is four orders of magnitude higher than 1a in the 1a k/o. The p-value is very high and non-significant.

A knockout does not necessarily manifest at the transcriptional level. It is not unusual to observe mRNA expression from mutant alleles. Irrespective of the transcript levels, at least with the antiserum that we have at our disposal, we observe no protein in Müller glia cells in our *rlbp1b*-KO fish.

9) In Figure 2, there is considerable variability in the measurements of retinoid content in WT adult animals. Between panels A, C and E, the mean WT levels differ markedly: for 11cisRAL (312, 177 and 280); for 11cisRE (12.7, 86.1 and 7.2); and for atRE (168, 460 and 187). These differences would benefit from explanation. Are these WT values significantly different from each other? Would it be better to use the WT grand means in the tests of significance for the knockouts?

This can be explained by technical variability due to (photo and thermal) isomerization during sample collection and processing, but also by the fact, that zebrafish and their eyes are growing throughout adulthood, so that the retina doesn’t reach a finite size with a precisely defined retinoid content at a specific age. Among other things, strain-to-strain variability could impact the extent of growth and therefore also the absolute amount of retinoids in the eye. Therefore, we felt that by using wild type siblings (or age-matched cousins in case of the double-KO) as controls, we could better control for age- and, importantly, strain-related variation and focus on the differences attributable to the genetic manipulation. Rather than comparing absolute amounts of retinoids in the adults between the different lines one could express the values as fold change or percentages of the wild type of each respective line. The results for the adults are presented in Author response image 2 as log2 fold change:

10) How critical is the role of Cralbpb in the MGC? The authors suggest that these cells could serve as a storage of 11-cis-retinoids. Could the authors elaborate a little this suggestion and how it might be explored? Along a similar line, which studies would the authors consider to investigate if 11cisRAL can be taken up by cones from the MGC (the question mark in the Graphical abstract)? In addition, if the MGC transform 11cisROL in 11cisRAL but the second is not transported back to the photoreceptors, what do the MGCs do with it? A storage for bright light conditions? It would be interesting to see the authors discuss this.

The precise role of Cralbpb in the MGC remains to be further investigated and we can only speculate on its function. In zebrafish, 11cisRE was mostly found in retinosomes in the RPE and was largely absent from MGC (see Babino et al., 2015). Thus, we conclude that MGCs do not store chromophore precursors in such form. However, MGCs express Cralbpb that can bind 11-cis-retinoids (e.g. produced by RGR?) and may deliver it to cone photoreceptors when needed. There is a whole host of novel gene targeting strategies available for zebrafish and mice. It will be fascinating to use this technology to unravel the role of Cralbp in MGCs and to elucidate the putative mechanism how cones benefit from this source for chromophore.

11) The morphology in Figure 5 (and the consecutive ones) suggests a stronger effect of the rlbp1a-KO in the ventral retina as compared to the dorsal. If this is the case, why could it be, and which implications could it have?

Photoreceptor degeneration does not occur uniformly in all parts of the retina. This has been reported in many mouse models of retinal degeneration. Whether this phenotype is related to differences in photoreceptor density and distribution and/or environmental factors such as light remains to be investigated. Additionally, we cannot exclude that the precise location and angle of the sections contribute to the apparent differences between ventral and dorsal regions in Figure 5, 6, and 7.

12) The first sentence of the discussion says "Our results showed that cones in the zebrafish retina rely on RPE-expressed Cralbpa for chromophore regeneration and retinal function" and summarizes the results, but as this has been extensively suggested, it would perhaps be better to focus more on the lack of effect of Cralbpb in MCG on chromophore regeneration. An important novelty is the use of a subfunctionalized teleost model to study RLBP1-KO related diseases.

We prefer to begin the discussion as it is. However, we now discuss the contribution of *rlbp1b* in a sentence further below in the discussion lines 299-300 in the revised manuscript:

“On the other hand, knockout of MGC-expressed rlbp1b in zebrafish did not affect 11cisRAL content, nor morphology and only marginally affected retinal function in the double-KO in bright light conditions.”

13) If cralbp1a is completely KO but there is still some chromophore regeneration, what explanation for this can the authors suggest?

This phenomenon is also known from mouse models and human patients, which exhibit prolonged, but not absent regeneration even though CRALBP is expected to be absent from both RPE and MGCs. It is also known, that retinoids can diffuse without the involvement of any retinoid binding proteins. Of course, their transport is greatly facilitated by proteins such as CRALBP, but not entirely dependent on them.

14) What do the authors speculate that could happen with the Retinomotor circadian movements? Would the Cralbp1a-KO fish have a similar response as for the dark-adapted retina? Could these movements be altered, therefore altering the circadian chromophore mobilisation and pigment regeneration and contributing to the accumulation of the small oil droplets/retinosomes in the RPE?

These are very interesting thoughts, but this seems beyond the scope of the current study.

15) For the insertion of the sgRNA and Cas-GFP, the authors used the injection methods. Did they try or consider using electroporation, working very well for isolated mammalian cells?

In zebrafish, injections in 1-cell stage embryos is an established technique that works very well in our hands. Electroporating zebrafish embryos is not quite as trivial as it might be for other cellular systems, at least in our experience.

16) In Figure 1A, the CRALBPas images are rather faint, probably due to poor contrast between background autofluorescence and the real signal, especially the two dbKO images. Presumably a negative control photo exists that could be added to help distinguish between the autofluorescence (due to the fixative, for example) and the real signal.

Indeed, formaldehyde fixed retinal tissue of fish has intrinsic autofluorescence. We always treated the controls and mutants with the same settings and adjustments during acquisition and processing of the images. Thus, the reviewers and readers of the paper should be able to appreciate the differences between the samples. However, we acknowledge that the settings might not have been optimal and the signal is relatively faint when we reduce the background signal as can be appreciated on the BandC adjusted Figure 1. We have attached another figure to this letter as an example showing a 5 dpf wild type and double-KO section (see Author response 3).

**Author response image 3. sa2fig3:** 

17) Figure 3: These 5 dpf b-wave signals seem huge, at ~10 mV. For comparison, Bilotta et al., (2001) reported ~50 µV for the b-wave in 8 dpf animals, so your values are 200x higher. If the scale was meant to show "µV" then the signals would be around 5x smaller than theirs. Ideally, the reader should have access to some samples of b-wave records (rather than just measured amplitudes), in order to judge the quality of the recordings, so please provide this. As a minimum, a panel of dark-adapted ERG traces should be presented as a Supplementary Figure.

We appreciate the suggestion to include dark-adapted ERG traces and have added a Supplementary figure with representative traces of the different mutant lines and noted this on lines 179-180. In our setup, we use the eye cup preparation which yields better signal compared to the original technique published by Bilotta et al., (2001), where measurements were performed in whole larvae. On top of that, we expect, that the precise configuration of the setup can account for some differences. Our measurements are, however, consistent with multiple publications from our lab in recent years.

18) Before the authors can state that "We observed no pronounced changes of cone responses in Cralbpb-deficient larval eyes" they need to illustrate rlbp1b KO responses in Figure 4. At present, the double KO plots in panel B2 ("BL", and "BL+reDA") appear to show considerably greater attenuation than for the rlbp1a KO plots in panel A2, strongly suggesting an effect of Cralbpb. To make the above statement, it is necessary to present the data.

This is a good point. However, we do not state that there is no effect, but that the effect is rather mild. Admittedly, we only measured ERG responses of *rlbp1b*-KO larvae in the dark-adapted state (we clarified this by adding “dark-adapted” to the statement above, line 210 of the revised manuscript), since we knew from the retinoid measurements in larvae, that 11cisRAL concentration is not affected by the mutation even after bright light exposure (BL and BL+reDA conditions). Therefore, we directly set out to investigate retinal function in the double mutant. As pointed out in question (3), the 11cisRAL that we measured could be both free or protein-bound. The possibility remains that our measurements of 11cisRAL were masked by RGR-bound retinoids. The latter effect may also explain the more severe phenotype in double-KO despite the apparently normal amounts of 11cisRAL in *rlbp1b*-KO. We added the following paragraph to the discussion lines 306-311 of the revised manuscript:

“Also, RGR in MGCs was implicated in light-dependent chromophore regeneration (Morshedian et al., 2019) and Cralbpb may drive this reaction during bright light conditions. In this line of thought our measurements of 11-*cis*-retinal in the *rlbp1b*-KO could be masked by RGR-bound retinoids and provide an explanation for the more severe phenotype in in double-KO, despite the apparently normal amounts of 11-*cis*-retinal in *rlbp1b*-KO.”

19) The authors used CRALBP antiserum to detect both Cralbpa and Cralbpb. Presumably it is difficult to generate specific antibodies if the two proteins are highly identical. Instead ISH with riboprobes specific against each paralog could be used. If the coding regions are too similar,the 3´UTR could be used. The authors have surely considered these options so a comment would be appropriate.

A gene knockout does not necessarily result in mRNA decay and an in situ hybridization with labelled ribo-probes would not be an appropriate method to confirm it.

20) The option of generating a dKO Cralbp1a/RPE65 would be an interesting next step continuing with this research line to investigate correlation between the RPE65 genes (the are three paralogs) and rlpb1a. This could be interesting to see a key protein not in transporting but to convert all-trans retinal back to 11-cis retinal, widening the spectrum of diseases to be analysed.

We agree, this is a fantastic idea and we hope that such a mutant fish line will be soon available.

21) Discussion mentions (line 495 onwards) studies that use single-cell transcriptomics. The authors are encouraged to embrace this option in the near future. It would be interesting to see the transcriptome also of the KOs with spatial resolution, and using ISS or SCRINSHOT spatial mapping for specific transcripts in intact sections in situ.

Again, this is a promising approach and we will apply it if we can secure enough funding to continue our project.